# ACTIVE PROMPTING WITH CHAIN-OF-THOUGHT FOR LARGE LANGUAGE MODELS

## ABSTRACT

The increasing scale of large language models (LLMs) brings emergent abilities to various complex tasks requiring reasoning, such as arithmetic and commonsense reasoning. It is known that the effective design of task-specific prompts is critical for LLMs' ability to produce high-quality answers. In particular, an effective approach for complex question-and-answering tasks is example-based prompting with chain-of-thought (CoT) reasoning, which significantly improves the performance of LLMs. However, current CoT methods rely on a fixed set of human-annotated exemplars, which are not necessarily the most effective examples for different tasks. This paper proposes a new method, **Active-Prompt**, to adapt LLMs to different tasks with task-specific example prompts (annotated with human-designed CoT reasoning). For this purpose, we propose a solution to the key problem of determining which questions are the most important and helpful to annotate from a pool of task-specific queries. By borrowing ideas from the related problem of uncertainty-based active learning, we introduce several metrics to characterize the uncertainty so as to select the most uncertain questions for annotation. Experimental results demonstrate the superiority of our proposed method, achieving superior performance on eight complex reasoning tasks. Further analyses of different uncertainty metrics, pool sizes, zero-shot learning, and accuracy-uncertainty relationships demonstrate the effectiveness of our method.[1]

## 1 INTRODUCTION

Large language models (LLMs) (Raffel et al., 2020; Brown et al., 2020; Chowdhery et al., 2022; Zhang et al., 2022a; Tay et al., 2022; Scao et al., 2022; Zeng et al., 2022; Smith et al., 2022) have achieved great success in recent years. A typical way of applying LLMs is in-context learning (Brown et al., 2020) by providing a number of instructions and exemplars, which performs well on conventional language understanding and generation tasks but performs poorly on complex reasoning tasks (Rae et al., 2021; Liang et al., 2022; Wei et al., 2022a). Recent prompting studies (Wei et al., 2022b; Wang et al., 2022c; Zhou et al., 2022) found that elaborating the reasoning steps in the exemplars endows LLMs with good reasoning abilities, namely chain-of-thought (CoT) prompting. However, chain-of-thought prompting depends on human engineering: it requires humans to select a few informative questions and then annotate them with CoT and answers. The human-annotated exemplars (questions with annotated CoT and answers) are not necessarily the most effective for different tasks. For example, the original chain-of-thought prompting (Wei et al., 2022b) crafted exemplars for eight questions, which are either randomly selected from the training set or manually composed by humans. Due to there being a significant variance in the nature of reasoning tasks in terms of difficulty, scope, domain, and so on, we do not know what kind of question is the most worthy of annotating. It is also not clear whether a particular set of exemplars are the best to elicit the desired information. However, the good news is that annotating eight exemplars for different tasks is trivial. It costs little money and human effort. In light of this, we identify the key problem as to how to determine which questions are the most important and helpful for annotation. We propose a solution to this problem by leveraging uncertainty and introducing a bit human effort to annotate a small set of questions. The annotation budget is reasonable.

---

[1]Our code will be released in the final version.

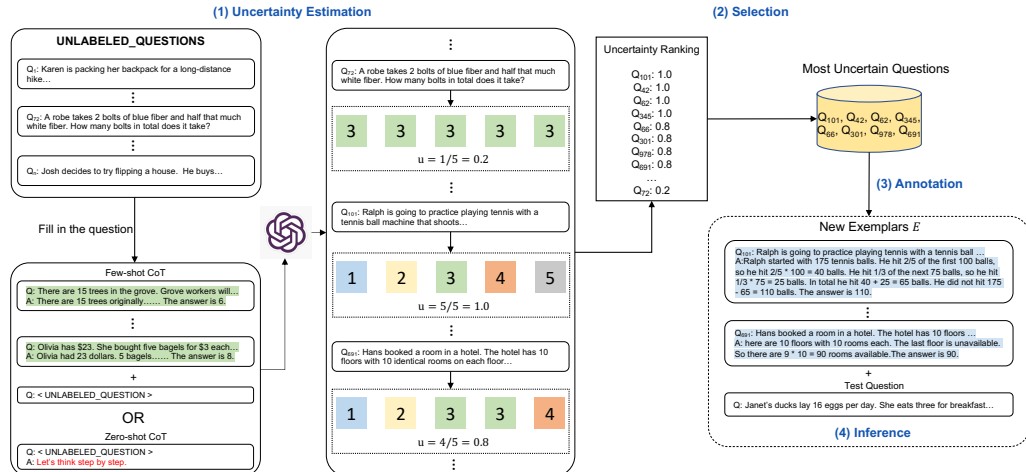

Figure 1: Illustrations of our proposed approach. There are four stages. **(1) Uncertainty Estimation**: with or without a few human-written chain-of-thoughts, we query the large language model $k$ ($k = 5$ in this illustration) times to generate possible answers with intermediate steps for a set of training questions. Then we calculate the uncertainty $u$ based on $k$ answers via an uncertainty metric (we use disagreement in this illustration). **(2) Selection**: according to the uncertainty, we select the most uncertain questions for annotation. **(3) Annotation**: we involve humans to annotate the selected questions. **(4) Inference**: infer each question with the new annotated exemplars.

By borrowing ideas from the related problem of uncertainty-based active learning (Gentile et al., 2022), we introduce several metrics to characterize the uncertainty among the model's predictions on each question. Therefore, we propose a new uncertainty-based annotation strategy that chooses a number of questions from the downstream dataset and involves humans annotating the rational chains, significantly improving the performance. Specifically, given a dataset $D$, we first ask the model to answer it $k$ times. Then we calculate the uncertainty $u$ of this model based on $k$ answers to each question. With $u$, we select the most uncertain $n$ questions with the largest $u$ and annotate these questions by the oracle to craft new exemplars $E$. Finally, we pre-pend $E$ to each test question following the standard recipe of chain-of-thought prompting (Wei et al., 2022b). The schematics of our proposed approach are illustrated in Figure 1. There are several different ways for uncertainty estimation in the literature (Settles, 2009; Culotta & McCallum, 2005). In our main experiments, we characterize the uncertainty $u$ by the disagreement and entropy of all predicted answers. In addition, we also investigate other different uncertainty metrics, like variance and self-confidence. For self-confidence, we re-organize the generated answer with the question by a new template and then ask the model's confidence for such generation. In this scenario, $u$ is defined as a categorical variable from {very confident, confident, not confident, wrong answer}. It is observed that the disagreement, entropy, and variance perform similarly well, while self-confidence is not working because LLMs are prone to be over-confident.

We conduct our experiments on eight datasets, spanning arithmetic reasoning, commonsense reasoning, and symbolic reasoning. Experimental results demonstrate the effectiveness of our proposed method by outperforming the competitive baseline models on them. Further analyses of different uncertainty metrics, pool sizes, zero-shot learning, and accuracy-uncertainty relationship display the benefits of each proposed module and reveal their effects. Our contributions are threefold: 1) We propose to judiciously select the most helpful and informative questions for annotation, reducing the human engineering workload. 2) We introduce an effective uncertainty-based question selection strategy with several different uncertainty metrics. 3) Our proposed method surpasses competitive baseline models by a large margin on multiple reasoning tasks. To the best of our knowledge, our work is the first to demonstrate the benefits of active question selection in chain-of-thought prompting for solving complex reasoning tasks.

## 2 ACTIVE-PROMPT

The schematic illustrations of our proposed approach are illustrated in Figure 1. Given $l$ unlabeled training data $D_{tr} = \{q_1, q_2, ..., q_l\}$ and $m$ test data $D_{te} = \{p_1, p_2, ..., p_m\}$ with each

$q$ and $p$ indicating the question without any answer or reasoning steps, our goal is to annotate only $n$ questions from $D_{tr}$ as few-shot exemplars by constructing a new exemplar set $E = \{(q_1, c_1, a_1), (q_2, c_2, a_2), ..., (q_n, c_n, a_n)\}$ with reasoning steps $c$ and the answer $a$. Then we use $E$ to prompt the test data $D_{te}$ and obtain the predictions. In this section, we explain how to select the $n$ most uncertain questions and annotate them.

## 2.1 UNCERTAINTY ESTIMATION

To select a few questions from a large dataset, we need an unsupervised method. Previous studies (Gentile et al., 2022) demonstrate that reducing the model's uncertainty helps improve the model's performance. Therefore, we introduce the uncertainty of LLMs as a metric to select data. In the chain-of-thought setting, we first forward the LLM $k$ times to obtain $k$ answers for each question. Then the uncertainty of a question could be measured in different ways. In our work, we consider four potential uncertainty metrics, described below.

**Disagreement** First, we consider measuring the uncertainty using the disagreement among $k$ generated answers $A = \{a_1, a_2, ..., a_k\}$. The disagreement is calculating the unique answers in the predictions. The implementation is simple. We first count the unique answers by a set operation to remove duplicate items, obtaining $h$ unique items $A = \{a_1, a_2, ..., a_h\}$. Then the disagreement is calculated by $u = h/k$.

**Entropy** The uncertainty could also be characterized by entropy, which is calculated by

$$u = \arg\max_i - \sum_{j=1}^{k} P_\theta(a_j|q_i) \ln P_\theta(a_j|q_i),\tag{1}$$

where $P_\theta(a_j|q_i)$ is the frequency of a certain predicted answer among all predictions. A larger entropy denotes greater uncertainty in the system, and a smaller entropy denotes smaller uncertainty. Therefore, in complex reasoning, questions with a relatively large entropy will be selected as candidates.

**Variance** We further consider variance as a kind of uncertainty metric, which we hypothesize might be more suitable for arithmetic answers.

$$u = \arg\max_i \frac{\sum_{j=1}^{k}(a_j - \bar{a})^2}{k-1}\Bigg|_{q=q_i},\tag{2}$$

where $\bar{a} = \frac{1}{k}\sum_{j=1}^{k} a_j$. It is observed that there is a huge variation in predicted answers. Some predicted answers are small numbers (e.g., 1), while some are large numbers (e.g., 10000). To mitigate the domination issue of large numbers, we propose to normalize the predictions by all the mentioned numbers in the question. For example, given a question *There are $x_1$ people. Each person has $x_2$ apples. How many apples are there altogether?* and a predicted answer $\hat{y}$, we obtain $\hat{y}/(|x_1| + |x_2|)$ after normalization.

**Self-Confidence** Estimating the uncertainty can also be achieved by the LLMs themselves, namely self-confidence. It can be obtained by querying the model with a manually crafted template $T$ like *For the question $q$ and the predicted answer $a$, report the confidence about the answer from choices. (a) very confident (b) confident (c) not confident (d) wrong answer.* Then we select the least confident questions by:

$$\begin{aligned} u &= \arg\max_i(1 - \max_j P_\theta(a_j|q_i)) \\ &= \arg\min_i \max_j P_\theta(a_j|q_i), \end{aligned}\tag{3}$$

where $P_\theta(a_j|q_i)$ is a categorical distribution from a set {very confident, confident, not confident, wrong answer}.

We first conduct a pilot study and find that disagreement-, entropy- and variance-based metrics perform competitively well, significantly outperforming self-confidence (Details are shown in Section 5.1). Therefore, in our experiments, we mainly apply disagreement and entropy for our approach, which are simple to implement.

## 2.2 Selection and Annotation

After obtaining the uncertainty of each question, we can establish an uncertainty ranking according to the uncertainty of each question. Then we will select the top-$n$ uncertain questions for annotation. If there are more than $n$ questions with the largest uncertainty, we will randomly select $n$ questions from them. These $n$ questions will be annotated with rationale chains and answers by human annotators to construct new exemplars $E = \{(q_1, c_1, a_1), ..., (q_n, c_n, a_n)\}$. $E$ will replace the initial $\hat{E}$ and we will use it for few-shot chain-to-thought prompting.

## 2.3 Inference

With the new annotated exemplars $E$, we prompt each question with them in the inference stage. In addition, we apply self-consistency (Wang et al., 2022c) to infer a question $m$ times with a temperature $T$, and then select the most consistent answer.

## 3 Experimental Settings

In this section, we describe the details of the datasets and evaluation metrics, the baseline models, and the implementation in the following three subsections. More details are included in Appendix A.

### 3.1 Datasets and Evaluation Metrics

Following the standard evaluation settings in LLMs reasoning studies (Wei et al., 2022b), our experiments are conducted on three types of datasets: GSM8K (Cobbe et al., 2021), ASDiv (Miao et al., 2020), SVAMP (Patel et al., 2021), AQuA (Ling et al., 2017), SingleEq (Koncel-Kedziorski et al., 2016), CSQA (Talmor et al., 2019), StrategyQA (Geva et al., 2021), and last letter concatenation (Wei et al., 2022b). For last letter concatenation, we test on an out-of-distribution setting, where the prompts are 2 letters while the test questions are 4 letters. The statistics of these datasets are reported in Table 4. We report the exact match accuracy as the evaluation metric.

### 3.2 Baselines

In our experiments, the following four methods serve as the main baselines: Chain-of-thought (CoT) (Wei et al., 2022b), Self-consistency (SC) (Wang et al., 2022c), Auto-CoT (Zhang et al., 2022b), and Random-CoT. Random-CoT shares the same annotation process as Active-Prompt. The only difference is that it randomly samples questions from the training data for annotation instead of applying our proposed uncertainty metrics. Our experiments are mainly based on CodeX `code-davinci-002` (Chen et al., 2021) for two reasons. First, it is the most capable model available at the time we were conducting our experiments, consistent with the observations in previous studies (Wei et al., 2022b; Wang et al., 2022c; Miao et al., 2020). Second, it is free of charge in the initial limited beta period. In addition to `code-davinci-002`, we also test the performance with `text-davinci-002`, `text-davinci-003` and `gpt-3.5-turbo` to verify our method's effectiveness in the main experiment. We call the APIs from OpenAI's services[2].

### 3.3 Implementation

**Hyperparameters**   In our implementation, the model could only access the training data $D = \{X_{tr}, Y_{tr}\}$ before inference, and is evaluated on the test data $D = \{X_{te}, Y_{te}\}$. We apply the same number of exemplars as Wei et al. (2022b), which is 8 for GSM8K, ASDiv, SVAMP, and SingleEq, 7 for CSQA, 6 for StrategyQA, 4 for AQuA and Letter (4). Given that some datasets (i.e., ASDiv, SVAMP, and SingleEq) only have the test split, we adopt the annotation result of GSM8K and transfer it to these datasets for inference. The transfer details are in Table 4. In the inference stage, we set temperature $T = 0.7$ and infer 40 times for each question. We then take the most consistent answer.

**Uncertainty Estimation**   At this stage, we start with a few manually annotated exemplars to help infer answers in the uncertainty estimation stage. These annotated exemplars are directly taken

---

[2]https://openai.com/api/

| METHOD | GSM8K | ASDIV | SVAMP | AQUA | SINGLEEQ | CSQA | STRATEGY | LETTER (4) | AVG. |
|---|---|---|---|---|---|---|---|---|---|
| Prior Best | $55.0^a$ | $75.3^b$ | $57.4^c$ | $37.9^d$ | $32.5^e$ | $91.2^f$ | $73.9^g$ | - | - |
| *text-davinci-002* | | | | | | | | | |
| Auto-CoT | 47.9 | - | 69.5 | 36.5 | 87.0 | 74.4 | 65.4 | 59.7 | - |
| CoT | 46.9 | 71.3 | 68.9 | 35.8 | 77.3 | 73.5 | 65.4 | 56.6 | 61.5 |
| SC | 58.2 | 76.9 | 78.2 | 41.8 | 87.2 | 72.9 | 70.7 | 57.6 | 67.9 |
| Random-CoT | 63.9 | 82.3 | 81.1 | 44.1 | 89.4 | 74.5 | 73.3 | 65.5 | 71.8 |
| Active-Prompt (D) | **73.2** | 83.2 | **82.7** | 48.4 | 90.6 | 76.6 | **76.9** | **67.7** | 74.9 |
| Active-Prompt (E) | 71.1 | **83.8** | 81.8 | **50.3** | **93.1** | **78.8** | **76.9** | 66.7 | **75.3** |
| *code-davinci-002* | | | | | | | | | |
| Auto-CoT | 62.8 | - | - | - | - | - | - | - | - |
| CoT | 63.1 | 80.4 | 76.4 | 45.3 | 93.1 | 77.9 | 73.2 | 70.4 | 72.5 |
| SC | 78.0 | 87.8 | 86.8 | 52.0 | 93.7 | 81.5 | 79.8 | 73.4 | 79.1 |
| Random-CoT | 78.6 | 87.1 | 88.0 | 53.1 | 94.0 | 82.1 | 79.4 | 73.3 | 79.4 |
| Active-Prompt (D) | 82.2 | 88.4 | **88.7** | 55.1 | 94.5 | **83.9** | **80.6** | 74.1 | 80.9 |
| Active-Prompt (E) | **83.4** | **89.3** | 87.5 | **57.0** | **95.5** | 82.6 | **80.6** | **76.7** | **81.6** |
| *text-davinci-003* | | | | | | | | | |
| CoT | 61.7 | 78.2 | 77.6 | 46.0 | 91.5 | 76.2 | 72.6 | 70.2 | 71.8 |
| Active-Prompt (D) | **65.6** | **79.8** | **80.5** | **48.0** | **93.1** | **78.9** | **74.2** | **71.2** | **73.9** |
| *gpt-3.5-turbo-0613* | | | | | | | | | |
| CoT | 74.2 | 82.5 | 83.8 | **50.0** | 95.0 | 79.9 | 80.5 | 82.0 | 78.5 |
| Active-Prompt (D) | **77.1** | **83.6** | **85.5** | **50.0** | **96.0** | **81.5** | **82.1** | **84.0** | **80.0** |

Table 1: The overall performance of Active-Prompt. **Bold** denotes the best result. $a$: Cobbe et al. (2021), $b$: Lan et al. (2022), $c$: Pi et al. (2022), $d$: Amini et al. (2019), $e$: Hu et al. (2019), $f$: Xu et al. (2021), $g$: Chowdhery et al. (2022). Statistics for CoT and SC mostly come from the original paper, with unreported entries sourced from DIVERSE (Li et al., 2023). All the results with *text-davinci-003* and *gpt-3.5-turbo* do not apply self-consistency considering the cost.

from Wei et al. (2022b). We call it the few-shot prompting trick to stabilize the prediction. However, our method is not dependent on few-shot prompting, other exemplar-free methods like zero-shot prompting (Kojima et al., 2022) could be applied, and we demonstrate that it works well in Section 5.1. In our experiments, we limit the size of candidate instances to 1,000. If the size of the original training data is larger than 1,000, we only randomly sample 1,000 instances from it and consider such a subset while estimating the uncertainty. If the size is smaller than 1,000, we will use the full data. We conducted the experiments with different pool sizes and found that 1,000 provides a robust performance and the performance gains of increasing the pool size would converge. $k$ is set to 10 for all the datasets in our main experiments. The analysis of performance v.s. $k$ is discussed in Section 5.1. The results show that with the increase in pool size, the performance continues to increase and will converge at $k = 10$. For the uncertainty metrics, we mainly report the performance of the disagreement-based (Active-Prompt (D)) and entropy-based (Active-Prompt (E)) methods. Due to it having been observed that StrategyQA often ties with the maximum disagreement to be 2/2 = 1, we also take the frequency into consideration for Active-Prompt (D).

**Annotation** Our approach needs human annotation for a few selected questions. The annotator is one of the co-authors and is familiar to machine learning and chain of thought prompting. Owing to the focus of our method being the example selection, rather than the annotation, the annotator did not do trial and error and conduct the minimum human engineering, referring to the previous annotation practices (Wei et al., 2022b). Given a question, the annotator would mainly write the reasoning steps and give the true answer to it. The effect of different annotators and the separate effects of selection and annotation are discussed in Sections 5.1.

## 4 EXPERIMENTAL RESULTS

The experimental results are displayed in Table 1. Overall, our model outperforms all baseline models by a large margin. Across eight benchmark datasets, Active-Prompt (D) achieves superior results

| METHOD | GSM8K | ASDiv | SingleEq | CSQA | Letter (4) |
|---|---|---|---|---|---|
| Auto-CoT | 62.8 | - | - | 74.4 | 59.7 |
| Manual-CoT | 63.1 | 80.4 | 87.5 | 73.5 | 56.6 |
| SC | 78.0 | 87.8 | 93.7 | 72.9 | 57.6 |
| Random-CoT | 78.6 | 87.1 | 94.0 | 74.5 | 65.5 |
| Zero-Shot-Active-Prompt | 82.2 | 86.7 | 94.2 | - | - |
| Active-Prompt (D) | 82.2 | 88.4 | 94.5 | 76.6 | **67.7** |
| Active-Prompt (E) | 83.4 | 89.3 | **95.5** | **78.8** | 66.7 |
| Active-Prompt (V) | 75.2 | 86.4 | 94.0 | - | - |
| Active-Prompt-Anno. (A) | 82.2 | 88.4 | 94.5 | 76.0 | 58.3 |
| Active-Prompt-Anno. (B) | **84.0** | **89.6** | 94.7 | 75.2 | 57.5 |

Table 2: Ablation study on three arithmetic reasoning tasks, CSQA, and Letter (4). Zero-Shot-Active-Prompt denotes the removal of the dependence of few-shot CoTs during uncertainty estimation. Anno. (A) and Anno. (B) are two different annotators. (D), (E), and (V) denote the disagreement, entropy, and variance, respectively. **Bold** represents the best among each dataset. The results of GSM8K, ASDiv, SingEq are obtained with `code-davinci-002` while the results of CSQA and Letter (4) are obtained with `text-davinci-002`.

with an average of 7.0% and 1.8% improvement over self-consistency with `text-davinci-002` and `code-davinci-002`, respectively. It demonstrates the effectiveness of our proposed active selection approach. In this section, we discuss the results of the arithmetic reasoning, commonsense and symbolic reasoning.

**Arithmetic Reasoning:**  Active-Prompt achieves the best performance compared with all baseline models, indicating the superiority of our method. Compared with the competitive baseline, self-consistency, Active-Prompt (D) outperforms it by an average of 2.1% with `code-davinci-002`. A larger improvement is observed with `text-davinci-002`, where Active-Prompt (D) improves over self-consistency by 7.2%. We notice that with `code-davinci-002`, the largest improvement is observed in GSM8K (4.2%) and AQuA (3.1%). One possible reason is that these two datasets do not require the transferability of CoT prompts because we can directly select and annotate the questions from their own training set. However, ASDiv, SVAMP and SingleEq do not have training data, so we need to transfer the annotated CoT from GSM8K to them. This suggests that how to better transfer prompts from one task to another is considered an important future research direction.

**Commonsense and Symbolic Reasoning:**  Consistent improvement is observed in commonsense reasoning and symbolic reasoning tasks. Active-Prompt outperforms self-consistency across all three tasks. Note that we test the out-of-distribution setting on Letter (4), which is more challenging, and Active-Prompt still achieves the best performance compared with all baseline models.

## 5 ANALYSIS

In this section, we further conduct several additional experiments to disclose the effects of few-shot prompts, the active selection, different annotators, uncertainty metrics, pool size, and prompt engineering. Finally, we analyze the relationship between uncertainty and accuracy, hoping to provide more explanation about how our method works.

### 5.1 ABLATION STUDY

In this section, we reveal the impact of various modules in our proposed model design. First, we reported the performance under the zero-shot setting by removing the dependency of a few exemplars, then explored the contributions of our proposed active example selection strategy. In addition, we explore the effects of different annotators, different uncertainty metrics, and pool sizes. To verify their contributions, we ablate them one by one and evaluate three downstream tasks: GSM8K, ASDiv, and SingleEq. The results are shown in Table 2.

**Effects of Few-Shot Prompts**  In our main experiments, we start with 4-8 manually annotated exemplars to help infer answers during the uncertainty estimation stage and demonstrate the effectiveness

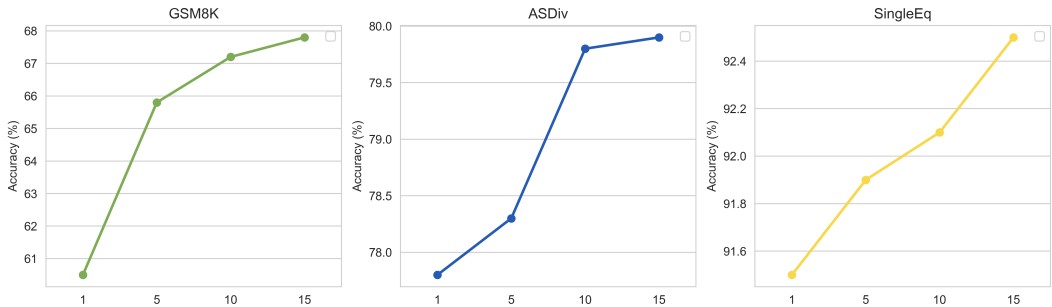

Figure 2: Comparison among the different numbers of predicted answers.

of our method. These annotated exemplars are directly taken from Wei et al. (2022b). However, our method is independent of these provided exemplars. In this section, we conduct further experiments with the assumption that we do not have access to them. Inspired by the recent research of Zero-Shot-CoT (Kojima et al., 2022), we found it is possible to bypass the manual effort of writing the initial exemplars. Instead of using 4-8 human-written exemplars to generate $k$ predictions, we simply add *"Let's think step by step."* and let LLMs generate the reasoning steps and the final answer. The results are shown in Table 2 Zero-Shot-Active-Prompt, which performs competitively to Active-Prompt, demonstrating that our method is not necessarily dependent on the few-shot exemplars.

**Effects of Active Selection** Our main contributions are the proposal of an effective example selection strategy (namely active selection). We replace the active selection with random selection by randomly selecting the same number of questions for annotation. The annotation process is exactly the same as Active-Prompt with the same annotation process and annotator. This model is called Random-CoT. The results are shown in Table 2. It is observed that Active-Prompt outperforms Random-CoT by a significant margin. Random-CoT only performs comparably to another baseline model self-consistency, illustrating that our applied annotation process has no advantages, and it is the active selection strategy that leads to performance gains. For example, on the GSM8K dataset, Random-CoT (78.6) slightly outperforms SC (78.0), while significantly underperforming Active-Prompt (82.2) by 3.6%. The full results of Random-CoT on all datasets are reported in Table 1 with a consistent performance drop compared with Active-Prompt.

**Effects of Annotators** In our main experiments, we asked the annotator not to do trial and error with minimum human engineering because the focus of our method is the question selection, rather than the best possible annotation. However, different annotators can still cause variations in the performance. In this section, we discuss the effects of different annotators. In addition to the annotator (annotator A), we directly use the human-annotated rationales from the GSM8K dataset (annotator B). The results are reported in Table 2. The results of annotators A and B are consistently better than baseline models, demonstrating the robustness of our proposed selection method. Surprisingly, we found that directly applying the solutions provided by GSM8K outperforms our annotated rationales, suggesting that the existing annotation of GSM8K is of high quality. In addition, we note that human prompt engineering has two complementary components: question selection and prompt template engineering. The method proposed in this work provides a good solution to the first problem. It is also possible to combine this technique with human-optimized prompt templates to further improve performance.

**Effects of Uncertainty Metrics** In our main experiments, we adopt disagreement as the uncertainty metric for most datasets and entropy for StrategyQA. In addition to those, other uncertainty metrics can be incorporated, which are investigated in our pilot study. In this section, we mainly discuss these four uncertainty metrics: disagreement, entropy, variance, and self-confidence. The definition of these four metrics is illustrated in Section 2.1. First, we found that disagreement is not applicable to datasets with limited search space. For example, the StrategyQA has only two labels (yes or no), and the predictions often tie in the maximum disagreement 2/2=1. Therefore, we adopt entropy for StrategyQA. Second, the self-confidence-based method performs badly, so we did not conduct experiments with it. We displayed an example of its prediction in Table 5. We conjecture that it is

| METHOD | GSM8K | CSQA | Letter (4) |
|---|---|---|---|
| TD-002 (CoT) | 46.9 | 73.5 | 56.6 |
| TD-002 → TD-002 (CoT) | 48.4 | 74.7 | 57.7 |
| TD-002 (SC) | 58.2 | 72.9 | 57.6 |
| CD-002 → TD-002 (SC) | 73.2 | 76.6 | 67.7 |
| TD-003 (CoT) | 61.7 | 76.2 | 70.2 |
| CD-002 → TD-003 (CoT) | 65.6 | 78.9 | 71.2 |
| TD-003 → TD-003 (CoT) | 67.2 | 80.8 | 73.7 |

Table 3: Analysis of the transferability of active exemplars. CD-002, TD-002, TD-003 denote `code-davinci-002`, `text-davinci-002`, and `text-davinci-003`. TD-002 (CoT), TD-002 (SC), and TD-003 (CoT) are three baseline methods without Active-Prompt. TD-002 → TD-002 (CoT) denotes selecting exemplars by `text-davinci-002` and inference by `text-davinci-002`. CD-002 → TD-002 (SC) denotes selecting exemplars by `code-davinci-002` and inference by `text-davinci-002`. CD-002 → TD-003 (CoT) denotes selecting exemplars by `code-davinci-002` and inference by `text-davinci-003`.

because GPT-3 is prone to be over-confident, which is consistent with previous observations (Si et al., 2022). Introducing an external well-trained discriminator to evaluate confidence is a practical way, and we leave it to future work. Last, the comparison between disagreement-, entropy- and variance-based methods are shown in Table 2. The results illustrate that they perform competitively well on ASDiv and SingleEq while disagreement and entropy outperform variance in GSM8K. Therefore, we simply choose disagreement and entropy as the primary metric in our main experiments.

**Effects of Pool Size**  In the first step for uncertainty estimation, we generate $k$ answers for each input question to construct a pool of predictions. Here, $k$ affects the performance of estimating the uncertainty, further affecting the downstream task's performance. To show the effect of the number of predicted answers, we plot the accuracy with respect to varying numbers of predicted answers (1, 5, 10, 15) in Figure 2 based on `text-davinci-003`. The results show that with the increase in pool size, the performance continues to increase and will converge at $k = 10$. It is intuitive that a small $k$ may confuse the selection process, leading to ties, while a large $k$ will lead to more accurate uncertainty estimation with better performance.

## 5.2 UNCERTAINTY ANALYSIS

The motivation of our proposed method is reducing the model's uncertainty to help elicit the reasoning ability of LLMs, further improving the few-shot prompting performance. In this section, we display the relationship between uncertainty and accuracy. In Appendix A Figure 3, we report the uncertainty quantity and accuracy on GSM8K, ASDiv, and SingleEq. We observe that there is a highly negative correlation between uncertainty and accuracy. With the decrease in uncertainty, the accuracy increases, demonstrating that reducing the model's uncertainty indeed helps improve the few-shot prompting-based predictions.

## 5.3 TRANSFERABILITY

In addressing the question of whether the uncertainty in selected exemplars is consistent across different models or if it originates from the specific task itself, an additional experiment was conducted. The experiment involves selecting exemplars using the `code-davinci-002` model and then performing inference using both `text-davinci-002` and `text-davinci-003` models. The underlying hypothesis is that if the uncertainty is inherent to the task, then the exemplars identified by Active-Prompt would exhibit transferability across models. In other words, the active exemplars identified by one model would be applicable and effective when transferred to other models. From the results in Table 3, it is observed that all three selection-based methods work well. The selected uncertain cases are related to tasks and can transfer to different models. It indicates that the uncertainty stems from the task and the exemplars identified by Active-Prompt have a good transferability. The results of this experiment provide insights into the nature of uncertainty in model predictions and its potential sources.

# 6 RELATED WORK

In this section, we present a review of reasoning ability, chain-of-thought prompting, and active learning methods.

## 6.1 REASONING ABILITY

Reasoning ability is essential to humans and is highly desirable for machine learning models. It consists of various sub-skills including commonsense reasoning (Zellers et al., 2018; Talmor et al., 2019; Bhagavatula et al., 2019), numerical reasoning (Dua et al., 2019), arithmetic reasoning (Koncel-Kedziorski et al., 2015; Roy & Roth, 2016; Miao et al., 2020; Cobbe et al., 2021), logical reasoning (Yu et al., 2020), tabular reasoning (Zhu et al., 2021), and so on. Previous efforts in machine learning exploited symbolic systems (Mihaylov & Frank, 2018; Ding et al., 2019; Feng et al., 2020; Wang et al., 2022b;a) and pre-training strategies (Deng et al., 2021; Asai & Hajishirzi, 2020; Pi et al., 2022). Recently, large language models with chain-of-thought prompting (Wei et al., 2022b; Wang et al., 2022c; Zhou et al., 2022; Zhang et al., 2022b) have demonstrated promising reasoning abilities with appropriately designed prompts, achieving competitive performance on several benchmarks.

## 6.2 CHAIN-OF-THOUGHT PROMPTING

Chain-of-thought prompting elicits the reasoning abilities of large language models. The original idea proposed by Wei et al. (2022b) is to enrich the few-shot examples with reasoning steps, which greatly improve the performance on complex tasks. Following Wei et al. (2022b), many studies improve standard CoTs in terms of self-consistency (Wang et al., 2022c), least-to-most prompting (Zhou et al., 2022), dynamic least-to-most prompting (Drozdov et al., 2022), bootstrapping (Zelikman et al., 2022), self-training (Huang et al., 2022), and verifier (Li et al., 2022). These studies greatly improve the performance based on CoT on complex tasks while they are limited to a fixed set of exemplars. Compared with them, we propose annotating the most important task-specific questions for easy adaptation. Auto-CoT (Zhang et al., 2022b) clusters test questions according to the diversity and uses zero-shot prompting for answers. Unlike our method, it requires going through the test dataset, and our experiments show our superior performance over Auto-CoT. Note that both diversity and uncertainty are useful for selecting the most informative questions, and they are complementary. We consider the combination of diversity and uncertainty as an important future direction.

## 6.3 ACTIVE LEARNING

Our work is also relevant to active learning (Cohn et al., 1996; Olsson, 2009; Settles, 2009; Rotman & Reichart, 2022), which aims to improve data labeling efficiency by finding the most helpful unlabeled data to annotate with reasonable budgets. Recent studies (Schröder et al., 2022; Köksal et al., 2022) demonstrate the benefits of active learning based approaches for fine-tuning large language models for classification tasks. Following this, we incorporate max-entropy (Roy & McCallum, 2001), and least confidence (Culotta & McCallum, 2005) algorithms into in-context learning scenarios, and we verify the effectiveness of chain-of-thought prompting especially for complex reasoning tasks.

# 7 CONCLUSION

In this paper, we proposed Active-Prompt to elicit reasoning abilities in large language models (LLMs). Inspired by the idea of annotating reasoning steps to obtain effective exemplars, we aim to select the most helpful questions for annotation judiciously instead of arbitrarily. For this purpose, we propose an uncertainty-based active selection strategy to determine which questions are the most important and helpful to annotate from a pool of task-specific questions. We introduce four different strategies of uncertainty estimation for Active-Prompt: disagreement, entropy, variance, and self-confidence. These four strategies characterize uncertainty from different perspectives and we primarily apply disagreement and entropy. Empirically, Active-Prompt achieved a promising performance on eight widely used datasets for arithmetic reasoning, commonsense reasoning, and symbolic reasoning. Further analyses of different uncertainty metrics, annotators, pool sizes, zero-shot learning, and an accuracy-uncertainty relationship demonstrate the effectiveness of our method.

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

# A  EXPERIMENTAL SETTINGS

In this section, we describe the details of the datasets and evaluation metrics, the baseline models, and the implementation in the following three subsections.

## A.1  DATASETS AND EVALUATION METRICS

Following the standard evaluation settings in LLMs reasoning studies (Wei et al., 2022b), our experiments are conducted on three types of datasets:

- Arithmetic Reasoning: GSM8K (Cobbe et al., 2021), ASDiv (Miao et al., 2020), and SVAMP (Patel et al., 2021), AQuA (Ling et al., 2017), and SingleEq (Koncel-Kedziorski et al., 2016).
- Commonsense Reasoning: CSQA (Talmor et al., 2019) and StrategyQA (Geva et al., 2021).
- Symbolic Reasoning: last letter concatenation (Wei et al., 2022b). This task evaluates the model's ability to concatenate the last letters of the words in a name. The standard in-distribution setting is trivial and previous methods have achieved almost 100% accuracy (Wei et al., 2022b). We test on an out-of-distribution setting, where the prompts are 2 letters while the test questions are 4 letters.

The statistics of these datasets are reported in Table 4. Note that in our experiments, we randomly sample 1000 data from the training set to reduce the computational cost. This may affect the performance of the uncertainty estimation. Intuitively, more training data will help capture the data distribution, leading to more precise uncertainty estimation. Given more financial support, the performance of our model will continue to increase. To make a fair comparison, we use the same test set as Wei et al. (2022b). We report the exact match accuracy as the evaluation metric.

| DATASET | TASK TYPE | # EX. | # TRAIN | # TEST | EVAL. SPLIT | TRANS. |
|---|---|---|---|---|---|---|
| GSM8K (Cobbe et al., 2021) | Arithmetic | 8 | 7473 | 1319 | Test | ✗ |
| ASDiv (Miao et al., 2020) | Arithmetic | 8 | - | 2096 | Test | ✓ |
| SVAMP (Patel et al., 2021) | Arithmetic | 8 | - | 1000 | Test | ✓ |
| AQuA (Ling et al., 2017) | Arithmetic | 4 | 97467 | 254 | Test | ✗ |
| SingleEq (Koncel-Kedziorski et al., 2016) | Arithmetic | 8 | - | 508 | Test | ✓ |
| CSQA* (Talmor et al., 2019) | Commonsense | 7 | 9741 | 1221 | Dev | ✗ |
| StrategyQA* (Geva et al., 2021) | Commonsense | 6 | 2061 | 1880 | Dev | ✗ |
| Letter (4) (Wei et al., 2022b) | Symbolic | 4 | 1000 | 1000 | Test (OOD) | ✗ |

Table 4: The statistics of the datasets used in this paper. # EX. are the number of few-shot chain-of-thought exemplars used to prompt each task in evaluation. # TRAIN and # TEST denote the number of training data and test data, respectively. Note that in our experiments, we randomly sample 1000 data from the training set to reduce the computational cost and use the same test set as Wei et al. (2022b). TRANS.: A checkmark denotes that the exemplars are from other datasets and then transferred to this task. *: CSQA and StrategyQA do not have publicly available test set labels, so we simply follow the setting by Wei et al. (2022b) to evaluate the performance of the development set.

## A.2  BASELINES

In our experiments, the following four methods serve as the main baselines:

- Chain-of-thought (CoT) (Wei et al., 2022b): standard chain-of-thought prompting which provides four to eight human-written exemplars consisting of a series of intermediate reasoning steps.
- Self-consistency (SC) (Wang et al., 2022c): an improved version of CoT. Instead of greedy decoding, it samples a set of reasoning paths and chooses the most common answer.
- Auto-CoT (Zhang et al., 2022b): an automatic exemplar construction method by clustering and generating rationales with zero-shot prompting (Kojima et al., 2022).
- Random-CoT: a baseline of Active-Prompt. It shares the same annotation process as Active-Prompt. The only difference is that it randomly samples questions from the training data for annotation instead of applying our proposed uncertainty metrics.

Our experiments are mainly based on CodeX `code-davinci-002` (Chen et al., 2021) for two reasons. First, it is the most capable model available at the time we were conducting our experiments, consistent with the observations in previous studies (Wei et al., 2022b; Wang et al., 2022c; Miao et al., 2020). Second, it is free of charge in the initial limited beta period. In addition to `code-davinci-002`, we

also test the performance with `text-davinci-002` and `text-davinci-003` to verify our method's effectiveness in the main experiment. We call the APIs directly from OpenAI's services[3].

### A.3 IMPLEMENTATION

**Hyperparameters** In our implementation, the model could only access the training data $D = \{X_{tr}, Y_{tr}\}$ before inference, and is evaluated on the test data $D = \{X_{te}, Y_{te}\}$. We apply the same number of exemplars as Wei et al. (2022b), which is 8 for GSM8K, ASDiv, SVAMP, and SingleEq, 7 for CSQA, 6 for StrategyQA, 4 for AQuA and Letter (4). Given that some datasets (i.e., ASDiv, SVAMP, and SingleEq) only have the test split, we adopt the annotation result of GSM8K and transfer it to these datasets for inference. The transfer details are in Table 4. In the inference stage, we set temperature $T = 0.7$ and infer 40 times for each question. We then take the most consistent answer.

**Uncertainty Estimation** At this stage, we start with a few manually annotated exemplars to help infer answers in the uncertainty estimation stage. These annotated exemplars are directly taken from Wei et al. (2022b). We call it few-shot prompting trick to stabilize the prediction. However, our method is not dependent on few-shot prompting, other exemplar-free methods like zero-shot prompting (Kojima et al., 2022) could be applied, and we demonstrate that it works well in Section 5.1. For the uncertainty metrics, we mainly report the performance of the disagreement-based (Active-Prompt (D)) and entropy-based (Active-Prompt (E)) methods. Due to it having been observed that StrategyQA often ties with the maximum disagreement to be 2/2 = 1, we also take the frequency into consideration for Active-Prompt (D).

**Annotation** Our approach needs human annotation for a few selected questions. The annotator is one of the co-authors and is familiar to machine learning and chain of thought prompting. Owing to the focus of our method being the example selection, rather than the annotation, the annotator did not do trial and error and conduct the minimum human engineering, referring to the previous annotation practices (Wei et al., 2022b). Given a question, the annotator would mainly write the reasoning steps and give the true answer to it. The effect of different annotators and the separate effects of selection and annotation are discussed in Sections 5.1.

## B UNCERTAINTY ANALYSIS

Figure 3 shows the relation between accuracy and uncertainty.

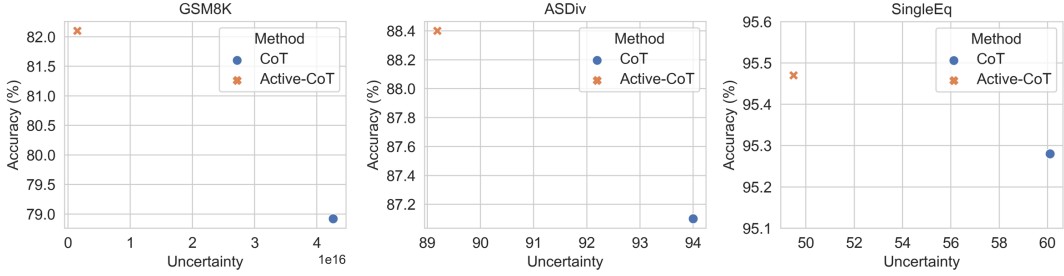

Figure 3: The relation between uncertainty and accuracy.

---
[3]https://openai.com/api/

## C  CASE STUDY OF CONFIDENCE-BASED METHOD

| | |
|---|---|
| Question | Every 2 miles a car drives the tires rotate 725 times. Jeremy drives 400 miles a month. If each tire can undergo 10,440,000 rotations how many years before the tire needs to be replaced? |
| Predicted Answer | Each tire can undergo 10,440,000 rotations. In 400 miles, the car travels 400 / 2 = 200 miles. In 200 miles, each tire rotates 200 x 725 = 145,000 times. 145,000 / 10,440,000 = 0.0139. The answer is 0.0139 years. |
| Confidence | Report confidence about the answer from choices. (a) very confident (b) confident (c) not confident (d) wrong answer The answer is calculated by the number of rotations for each tire for a month and divide by the maximum number of rotations it can do, which is 145,000/10,440,000=0.0139. The logic is correct and no calculation error. So, choose (a). I am very confident. |
| Label | 6 |

Table 5: An example of self-confidence-based prompting process and the results.

## D  COMPARISON WITH DIVERSITY-BASED METHODS

| METHOD | GSM8K | MultiArith | AddSub |
|---|---|---|---|
| Auto-CoT | 62.8 | 93.2 | 91.9 |
| Active-Prompt | **67.0** | **95.5** | **93.2** |

Table 6: Comparison with Auto-CoT. The results of Auto-CoT are taken directly from the original paper. For a fair comparison, none of the results apply the self-consistency method. Active-Prompt applies the rationales annotated by humans. **Bold** represents the best among each dataset. All the results are obtained with `code-davinci-002`.

Auto-CoT (Zhang et al., 2022b) proposes a diversity-based method for question selection, and ours proposes an uncertainty-based method for it. In this section, we compare our method with Auto-CoT to demonstrate their effectiveness and differences. Owing to Auto-CoT only reporting the results on GSM8K, MultiArith, and AddSub on `code-davinci-002` without self-consistency, we first compare our method with it on these three datasets in the same setting. The results are shown in Table 6. It is observed that Active-Prompt outperforms Auto-CoT by a large margin. We attribute the improvement to uncertainty-based selection and human annotation. Note that both diversity and uncertainty are useful for selecting the most informative questions, and they are complementary. We consider the combination of diversity and uncertainty as an important future direction.

## E  COMPARISON WITH COMPLEX-COT

| METHOD | GSM8K | AsDiv | SVAMP | SingleEq |
|---|---|---|---|---|
| Complex-CoT | 76.3 | 82.4 | 79.9 | 93.3 |
| Active-Prompt (D) | **77.1** | **83.6** | **85.5** | **96.0** |

Table 7: Comparison with Complex-CoT. **Bold** represents the best among each dataset. All the results are obtained with `gpt-3.5-turbo`.

Complex-CoT (Fu et al., 2022) is a strong baseline which takes the complexity of prompts into consideration and proposes to select those complex prompts as exemplars. We find that Active-Prompt outperforms Complex-CoT, demonstrating the effectiveness of our proposed uncertainty-based methods. In addition, we can combine the uncertainty and the complexity together to achieve a better performance, and we leave this for future work.

## F    PERFORMANCE OF WEAKER MODELS

Our main experiments are conducted based on GPT-series models which are powerful. One may wonder about the performance of weaker / smaller models, e.g., Llama-series models (Touvron et al., 2023a;b). In this section, we investigate the effectiveness of Active-Prompt with Llama-2 models and the results are shown below.

| METHOD | GSM8K | AsDiv | SVAMP | SingleEq |
|---|---|---|---|---|
| CoT (Llama2-13b-chat) | 29.1 | 57.4 | 45.7 | 75.4 |
| Active-Prompt (Llama2-13b-chat) | 27.2 | 53.9 | 47.4 | 68.2 |
| CoT (Llama2-70b-chat) | 54.8 | 73.2 | 77.4 | 84.6 |
| Active-Prompt (Llama2-70B-chat) | 56.9 | 73.0 | 82.5 | 83.2 |
| CoT (gpt-3.5-turbo) | 74.2 | 82.5 | 83.8 | 95.0 |
| Active-Prompt (gpt-3.5-turbo) | **77.1** | **83.6** | **85.5** | **96.0** |

Table 8: Comparison with weaker models. **Bold** represents the best among each dataset.

Firstly, because smaller models do not have a good chain-of-thought ability (Wei et al., 2022a;b), it is difficult for Llama2-13b-chat to understand the chain-of-thought prompting, with poor performance of both CoT and Active-Prompt. Secondly, our proposed Active-Prompt outperforms CoT by a large margin, demonstrating this method is still useful for weaker models. Note that we are using the instruction-tuned version of Llama2-70b in all our experiments (i.e., Lllama2-70b-chat) because it is able to understand complex chain-of-thought prompting and follow human instructions.

## G    TRANSFERABILITY BETWEEN GPT AND LLAMA MODELS

We also investigate the transferability between GPT and Llama models. Because smaller Llama models perform poorly on reasoning tasks, we conduct experiments with Llama2-70b-chat. We conduct two types of experiments: (1) select questions by gpt-3.5-turbo and infer by Llama2-70b-chat and (2) select questions by Llama2-70b-chat and infer by gpt-3.5-turbo. Note that we are using the 0613 version of gpt-3.5-turbo. The results are shown below. The model before the arrow denotes the model for actively selecting questions while the model after the arrow denotes the model for inference. The results demonstrate the feasibility of selecting questions with smaller models (Llama2-70b-chat) and applying the selected questions to larger models (gpt-3.5-turbo). In addition, selecting questions with larger models and applying them to smaller models has better performance.

| METHOD | GSM8K | AsDiv | SVAMP | SingleEq |
|---|---|---|---|---|
| gpt-3.5-turbo | 74.2 | 82.5 | 83.8 | 95.0 |
| gpt-3.5-turbo → gpt-3.5-turbo | 77.1 | 83.6 | 85.5 | 96.0 |
| Llama2-70b-chat → gpt-3.5-turbo | 78.7 | 85.9 | 83.1 | 91.5 |
| Llama2-70b-chat | 54.8 | 73.2 | 77.4 | 84.6 |
| Llama2-70b-chat → Llama2-70b-chat | 56.9 | 74.9 | 82.5 | 83.2 |
| gpt-3.5-turbo → Llama2-70b-chat | 58.9 | 74.7 | 81.2 | 86.0 |

Table 9: Transferability between gpt-3.5-turbo and Llama models.

## H    COSTS OF ACTIVE-PROMPT

Compared with selecting questions by humans, our proposed method is more efficient. For a new task, users need to do trials and errors a lot of times which costs a lot of human effort with unstable performance. Even so, the selected questions are still suboptimal. Second, as mentioned in Section 3.3, we limit the size of candidate instances to 1,000 which greatly reduces the cost. 1,000 is a good balance between cost and performance. We verified that with more than 1,000 instances, the performance would converge. Doing uncertainty estimation 10 times with a pool of 1,000 questions is acceptable. The cost is smaller than self-consistency, which usually requires 40 times inference, although it is an orthogonal technique and can be complementary to ours. In addition, inspired by the

new experimental results in Appendix G, we are excited to find that questions selected by smaller models (e.g., Llama) perform well with larger models (e.g., gpt-3.5-turbo). Considering models like Llama are open-source which does not cause API cost, one may use it (with GPU) to replace black-box API.

## I    ABLATION STUDY OF LONGER CoT ANNOTATIONS

Furthermore, we conduct an ablation study to differentiate the impacts of longer CoT annotations from our method. To explore this, we extended the length of the original CoT (Wei et al., 2022b) annotations to an average of 155 words, comparable to our average length of 160 words. The results are shown in Table 10. Our findings show that merely increasing the length of CoT annotations does not lead to improved performance, and in some cases, even reduces it. In contrast, our Active-Prompt method consistently demonstrates superior performance. This suggests that the selection of questions, rather than their length, contributes significantly to the improved results. Our approach effectively identifies and utilizes more informative examples for annotations.

| METHOD | GSM8K | AsDiv | SVAMP | SingleEq |
|---|---|---|---|---|
| gpt-3.5-turbo (original CoT) | 74.2 | 82.5 | 83.8 | 95.0 |
| gpt-3.5-turbo (longer CoT) | 69.4 | 69.2 | 70.4 | 83.2 |
| gpt-3.5-turbo (active-prompt) | 77.1 | 83.6 | 85.5 | 96.0 |

Table 10: Ablation Study of Longer CoT Annotations.

## J    FULL EXEMPLARS GENERATED BY ACTIVE-PROMPT

| Exemplars |
| --- |
| Q: As a freelancer, Baylor is paid for every finished work of a client he does on a freelance marketplace. Currently, he has $4000 on his dashboard from previous work done. He is currently working for three clients, with the first client paying him half the amount of money he currently has on his dashboard once the job is done. The second client will pay him 2/5 times more money than the first client once Baylor finishes his work. The third client will pay him twice the amount of money the first and second clients pay him together once he finishes the job. How much money will Baylor have in his dashboard after all the clients pay him for his work? |
| A: Let's think step by step. To calculate how much money will Baylor have in his leaderboard after all clients pay him for this work, we need to know the amount of payment from each client and add them together with original value on the leaderboard. The first client will pay Baylor 1/2*$4000=$2000. The second client will pay Baylor 2/5 times more money than the first client, a total of 2/5*$2000 = $800. The total amount of money the second client will pay Baylor once the job is done is $2000+$800 = $2800. The total amount of money the first and the second client will pay Baylor after their work is finished is $2800+$2000 = $4800. The third client will pay him twice the amount of money the first and second clients pay him together, a total of 2*$4800 = $9600. Together, the three clients will pay Baylor $9600+$4800 = $14400 once he finishes their jobs. In his dashboard, Baylor will have $14400+$4000 = $18400. The answer is 18400. |
| Q: Josh and Anna were both born on August 17th, but in different years. To consolidate celebrations they also got married on August 17 when Josh turned 22. If today they're celebrating 30 years of marriage and their combined age is exactly 5 times what Josh's age was when they married, how old was Anna when they got married? |
| A: Let's think step by step. To calculate how old was Anna when they got married, we have to know their combined age, Josh's age after 30 years, and Anna's age after 30 years from their marriage. Since their combined age is 5 times Josh's age when he got married, their combined age is 5 * 22 = 110 years. Josh must be 30 years older than his age when they got married, so he is 22 + 30 = 52 years old now. Therefore, Anna's current age will be 110 - 52 = 58 years. If they married 30 years ago, Anna must have been 58 - 30 = 28 years old when they married The answer is 28. |
| Q: Mary is trying to get Zoom set up for the first time. She spent 10 minutes downloading the Mac version, only to realize she needed the Windows version, which took three times as long to download. During her first call, the audio glitched twice for 4 minutes each time and the video glitched once for 6 minutes. She spent twice as long talking without glitches as with glitches. How much time did Mary spend downloading Zoom and talking in total? |
| A: Let's think step by step. To calculate how much time did Mary spend downloading Zoom and talking in total, we need to know the downloading time of the Mac version, downloading time of the Windows version, the total time of audio glitches and video glitches, and the total glitch-free time. The Windows version's downloading time is three times that of the Mac version, which is 3 * 10 = 30 minutes. The total time of audio glitches and video glitches is 4 minutes/glitch * 2 audio glitches + 6 minutes audio glitches = 14 minutes. The glitch-free time is twice as long as the time with glitch, which is 2 * 14 = 28 minutes. The total time Mary spent downloading Zoom and talking is to add all together, 10 minutes + 30 minutes + 14 minutes + 28 minutes = 82 minutes. The answer is 82. |
| Q: Valerie needs to put stamps on the envelopes she is about to mail. She has thank you cards for each of her grandmother, uncle and aunt for the birthday presents they sent. She also has to pay the water bill and the electric bill separately. She wants to send three more mail-in rebates than she does bills and she has twice as many job applications as rebates to mail. How many stamps does she need if everything needs 1 stamp except the electric bill, which needs 2? |
| A: Let's think step by step. To calculate how many stamps does Valerie needs to put on the envelopes she is about to mail, we need to know the number of thank you cards for birthday presents, the number of bills, the number of mail-in rebates, the number of job applications, and the total number of mails to send. Since there are three people to thank, the number of thank you cards is 3. There are two bills which are the water bill and the electric bill. The number of mail is 3 more than the number of bills, which is 3 + 2 = 5. The number of job applications is twice as many as main-in rebates, which is 2 * 5 = 10 applications. The total number of mails to send is 3 + 2 + 5 + 10 = 20 pieces of mail to send. Each mail needs 1 stamp but electric bills, which need 2 stamps. So she needs 20 + 1 = 21 stamps. The answer is 21. |
| Q: At the duck park, there are 25 mallard ducks and ten less than twice as many geese as ducks. Then, a small flock of 4 ducks arrived at the park and joined the birds already there. If five less than 15 geese leave the park, how many more geese than ducks remain at the park? |
| A: Let's think step by step. To calculate how many more geese than ducks remain at the park, we need to know the number of geese initially at the park, the number of ducks after 4 ducks arrived, the number of geese left, and the number of geese remains. The number of geese initially at the park is ten less than twice as many geese as ducks, which is 2*25 - 10 = 40 geese. The number of ducks in the park after 4 ducks arrived is 25 + 4 = 29 ducks. The number of geese left is 5 less than 15 geese which are 15 - 5 = 10 geese left. The number of geese remaining is 40 - 10 = 30 geese remaining. Therefore, there are 30 - 29 = 1 more geese than ducks remaining at the park. The answer is 1. |

Table 11: Exemplars for full chain of thought prompt selected and annotated from GSM8K. This set of exemplars is used by GSM8K, ASDiv, SVAMP, and SingleEq.

| Exemplars |
|---|
| Q: John decides to stop delivering the newspapers he is supposed to deliver and instead steals them to recycle them for cash. The Monday-Saturday papers weigh 8 ounces each. The Sunday paper weighs twice as much. He is supposed to deliver 250 papers a day. He doesn't deliver them for ten weeks. If one ton of paper recycles for $20, how much did he make? 
 A: Let's think step by step. To calculate how much did John make if one ton of paper was recycled for $20, we need to know the weight of a weekly delivery of the papers in pounds, the total weight of paper delivered in 10 weeks, and the total weight of paper in tons. The weekly delivery of a paper consists of 6 days of 8 ounces of paper and one day of 2*8 = 16 ounces of paper, so the weekly weight of paper is 6 * 8 + 16 = 64 ounces, which is 64/16=4 pounds. Since John needs to deliver 250 papers a day, the total weight of paper delivered in a week is 4 * 250 = 1000 pounds. The total weight of papers in 10 weeks is 10 * 1000 = 10000 pounds. The weight of paper in tons is 10000/2000 = 5 tons. So he makes 5 * 20 = $100 from recycling the paper. The answer is 100. |
| Q: John buys a chair. He then buys a table that is 3 times the price of the chair. Then, he buys a couch that is 5 times the price of the table. If John paid $380 for all these items, what is the price of the couch? 
 A: Let's think step by step. To calculate the price of the couch, we need to know the price of the chair, the price of the table, and the relation between the chair, table, couch, and total money paid. Let x be the price of the chair, 3*x be the price of the table, and 5*(3*x) = 15*x be the price of the couch. The relationship between the chair, table, couch, and the total price paid is x + 3*x + 15*x = $380, which is 19*x = 380, and x=20. The price of the couch is 15*x, which is 15*20 = $300. The answer is 300. |
| Q: A hay farmer harvested 560 bales of hay from 5 acres of grass per month last year. This year, he planted an additional 7 acres of grass. If the farmer also owns 9 horses and each horse consumes 3 bales of hay a day, how many bales of hay would the farmer have left by the end of December if he starts feeding them this year's hay beginning the first day of September? 
 A: Let's think step by step. To calculate how many bales of hay would the farmer have left by the end of December if he starts feeding the horse this year's hay beginning the first day of September, we need to know the number of bales of hay that can harvest from each acre of grass, the number of acres of grass the farmer has this year, the total number of bales of hay can harvest per month, the number of acres of grass the farmer has this year, the total number of bales of hay can harvest this year, the number of days to feed the horse from September to December, the number of bales of hay his house eats per day, and the total number of bales of hay his houses will eat. The number of bales of hay that can harvest from each acre of grass is 560/5 = 112 bales of hay each month. The number of acres of grass the farmer has this year is 7 + 5 = 12 acres of grass. The number of bales of hay that can harvest per month from the 12 acres of grass is 12 * 112 = 1344 bales of hay per month. The total number of bales of hay he can harvest this year is 1344 * 12 = 16128. The number of days to feed the horse from September to December is a total of 30 + 31 + 30 + 31 = 122 days. The number of bales of hay his horse eats each day is 3 * 9 = 27. The total number of bales of hay his horses consumed in 122 days is 27 * 122 = 3294. The total number of bales remaining is 16128-3294 = 12834. The answer is 12834. |

Table 12: (Cont.) Exemplars for full chain of thought prompt selected and annotated from GSM8K. This set of exemplars is used by GSM8K, ASDiv, SVAMP, and SingleEq.

| Exemplars |
| --- |

Q: As a bicycle salesperson, Norman earns a fixed salary of $20 per week plus $6 per bicycle for the first six bicycles he sells, $12 per bicycle for the next six bicycles he sells, and $18 per bicycle for every bicycle sold after the first 12. This week, Norman earned more than twice as much as he did last week. If he sold A bicycles last week and Y bicycles this week, which of the following statements must be true? I. y > 2x II. y > x III. y > 3 Answer Choices: (A) I only (B) II only (C) I and II (D) II and III (E) I, II, and III
A: Let's think step by step. This is kind of like doing your taxes, lets summarize the data in terms of bike sales and salary. 0 Bikes - $20. 6 Bikes - $56. 12 Bikes - $128. More than 12 bikes - 128 + 18*(a-12) where x is the total number of bikes sold (when a>12). A = cycles sold last week. AS = Salary last week. Y = cycles sole this week. YS = Salary this week. given YS>2AS. Let's test all statements with A=0;. AS = 20. YS >40. True when Y = 4. satisfies all the statements but we can only be sure of iii as Y will only increase from here. So iii must be true. Eliminate A,B,C. lets test all statements with A=1;. AS = 26. YS>52. True when Y=6. Still satisfies all the statements - Nothing achieved. lets test all statements with A=6. AS = 56. YS>112. True when Y = 11. This proves statement i wrong, hence eliminate E so we are left with D. The answer is D.

Q: If k is a non-negative integer and $15^{\hat{k}}$ is a divisor of 823,435 then $5^{\hat{k}}$ - $k^{\hat{5}}$ = Answer Choices: (A) 0 (B) 1 (C) 35 (D) 120 (E) 245
A: Let's think step by step. 8+2+3+4+3+5 = 25, so this number is not divisible by 3 and thus not divisible by 15. Therefore, k=0. $5^{\hat{k}}$ - $k^{\hat{5}}$ =1-0=1. The answer is B.

Q: For the past 3 weeks, Rikki has gone to the gym an average of 2 times per week. After 1 more week, the average number of times Rikki goes to the gym increases to 3 times per week. How many times did Rikki visit the gym during the most recent week? Answer Choices: (A) 7 (B) 3 (C) 6 (D) 2 (E) 4
A: Let's think step by step. Total times per week over 3 weeks = 3*2 = 6. Total times per week over 4 weeks = 4*3 = 12. Number of visits in most recent week = 12-6 = 6 visits. The answer is C.

Q: A clock is set right at 8 a.m. The clock gains 10 minutes in 24 hours. What will be the true time when the clock indicates 1 p.m. on the following day? Answer Choices: (A) 48 minutes past 12 (B) 48 minutes past 32 (C) 48 minutes past 19 (D) 48 minutes past 11 (E) 48 minutes past 16
A: Let's think step by step. Time from 8 a.m. on a day to 1 p.m. on the following day = 29 hours. 24 hours 10 minutes of this clock = 24 hours of the correct clock. Hours of this clock = 24 hours of the correct clock. 29 hours of this clock = hours of the correct clock = 28 hrs 48 minutes of the correct clock. Therefore, the correct time is 28 hrs 48 minutes after 8 a.m. This is 48 minutes past 12. The answer is A.

Table 13: Exemplars for full chain of thought prompt selected and annotated from AQuA.

| Exemplars |
|---|
| Q: Where would you find a sloth that is not afraid of being hunted? Answer Choices: (A) forest canopy (B) nature preserve (C) zoo (D) tropical rainforest (E) commercial
A: Let's think step by step. To answer the question of where you would find a sloth that is not afraid of being hunted, we need to know where sloths live and when sloths are not afraid of being hunted. Sloths live in places where their habitats are well protected. Sloths are not afraid of being hunted when hunting is illegal in their living places. Of the above choices, (B) nature preserve makes the most sense because it well protects sloths' habitats and forbids hunting. The answer is B. |
| Q: What gift do people claim that god has for people? Answer Choices: (A) bible (B) everywhere (C) judge people (D) care for people (E) work miracles
A: Let's think step by step. To answer the question of what gift do people claim that god has for people, we need to know what is a gift from god and what people cannot do without the gift from god. The gift from god is an ability that people normally cannot achieve. People normally cannot create miracles unless it is gifted by god. Of the above choices, (E) work miracles make the most sense because miracles refer to something that people cannot achieve unless it is gifted by god. The answer is E. |
| Q: What is probably not built out of a wooden rod? Answer Choices: (A) broom (B) broom closet (C) old clock (D) hardware store (E) old houses
A: Let's think step by step. To answer the question of what is probably not built out of a wooden rod, we need to know what is a wooden rod and what a wooden rod can do. A wooden rod is a stick made of wood of various sizes. A wooden rod can be used as a part of tools or furniture. Of the above choices, (E) old houses make the most sense because the wooden rod is too weak to support old houses. The answer is E. |
| Q: Where do you use condoms? Answer Choices: (A) supermarket (B) dick (C) bedroom (D) medicine chest (E) bed
A: Let's think step by step. To answer the question of where do you use condoms, we need to know when do people need condoms and where is the place people stay when they use condoms. People use condoms when they have sex. People have sex and use condoms in their bedrooms. Of the above choices, (C) bedroom makes the most sense because it is most polite and appropriate for people to have sex with condoms in the bedroom. The answer is C. |
| Q: What happens soon after Christmas? Answer Choices: (A) buy presents for others (B) summer (C) easter (D) see work (E) halloween
A: Let's think step by step. To answer the question of what happens soon after Christmas, we need to know what means soon after Christmas and what can be happened soon after Christmas. Soon after Christmas means some days after Christmas day. Normally another holiday happens soon after Christmas. Of the above choices, (C) easter makes the most sense because easter is a holiday with the closest date after Christmas day. The answer is C. |
| Q: Finding information requires what other than research materials? Answer Choices: (A) internet (B) ask questions (C) time (D) searching for (E) logic
A: Let's think step by step. To answer the question of finding information requires what other than research materials, we need to know what people do when they are finding information and what are research materials. People spend time looking up something with purpose is called finding information. Research materials normally refer to the stuff they are looking for and necessary tools such as the internet. Of the above choices, (C) time makes the most sense because people definitely need to spend time to find information. The answer is C. |
| Q: Before someone can adopt the parent must do what with their offspring? Answer Choices: (A) biological child (B) give away (C) birth (D) abandon (E) orphan
A: Let's think step by step. To answer the question of before someone can adopt the parent must do what with their offspring, we need to know what is the requirement to adopt and what is the legal action to do with their offspring. Normally people can adopt when they don't have offspring or give up having their offspring. It is legal to decide not to have babies, which is giving away their offspring. Of the above choices, (B) give away makes the most sense because people must give away their offspring to be able to adopt. The answer is B. |

Table 14: Exemplars for full chain of thought prompt selected and annotated from CommonsenseQA.

| Exemplars |
|---|
| Q: Is it possible to get killed walking to the Very Large Telescope?
A: Let's think step by step. The Very Large Telescope is in the Atacama Desert. The Atacama Desert is the driest hot desert in the world. Thus, it is possible to get killed walking in the Atacama desert. The answer is yes. |
| Q: Did Clark Gable marry more women once than Richard Burton?
A: Let's think step by step. Richard Burton was married to four different women. Clark Gable was married to five different women, one time each. Thus, Clark Gable married more women than Richard Burton. The answer is yes. |
| Q: Can a human heart last from NYC to Raleigh NC by Toyota Hiux?
A: Let's think step by step. Human hearts can last up to 6 hours outside the body. The distance from NYC to Raleigh, NC is 505 miles. The top speed of a Toyota Hilux is 105 miles per hour. The time needed to travel from NYC to Raleigh, NC is 505 miles / 105 miles per hour = 4.8 hours. Thus, the time needed to travel from NYC to Raleigh NC is less than 6 hours. The answer is yes. |
| Q: Is H's most common two letter pair partner a freebie in Wheel of Fortune bonus round?
A: Let's think step by step. H forms the most common two letter pair in the English language along with the letter T. The Wheel of Fortune bonus round gives the player six free letters: R, S, T, L, N, E. Thus, The letter T is inclued in the given six free letters in the Wheel of Fortune bonus round. The answer is yes. |
| Q: Could someone in Tokyo take a taxi to the The Metropolitan Museum of Art?
A: Let's think step by step. Tokyo is located in Japan. The Metropolitan Museum of Art is in the United States. Japan and the United States are separated by the Pacific Ocean. A taxi is not capable of travelling over the Pacific Ocean. Thus, someone in Tokyo cannot take a taxi to the The Metropolitan Museum of Art. The answer is no. |
| Q: Does Olympia Washington share name with Hephaestus's workshop location?
A: Let's think step by step. Olympia Washington is named after Mount Olympus, which is a mountain range in Washington named after the ancient Greek Mount Olympus. Hephaestus was the ancient Greek god of the forge and had a workshop on Mount Olympus. Thus, Olympia Washington share name with Hephaestus's workshop location The answer is yes. |

Table 15: Exemplars for full chain of thought prompt selected and annotated from StrategyQA.

| Exemplars |
|---|
| Q: Take the last letters of the words in "Willie Valeria Zane" and concatenate them.
A: Let's think step by step. The last letter of "Willie" is "e". The last letter of "Valeria" is "a". The last letter of "Zane" is "e". Concatenating them is "eae". The answer is eae. |
| Q: Take the last letters of the words in "Kylie Guille Lakshmi" and concatenate them.
A: Let's think step by step. The last letter of "Kylie" is "e". The last letter of "Guille" is "e". The last letter of "Lakshmi" is "i". Concatenating them is "eei". The answer is eei. |
| Q: Take the last letters of the words in "Carmelo Reinaldo Sade" and concatenate them.
A: Let's think step by step. The last letter of "Carmelo" is "o". The last letter of "Reinaldo" is "o". The last letter of "Sade" is "e". Concatenating them is "ooe". The answer is ooe. |
| Q: Take the last letters of the words in "Gabe Ventura Woody" and concatenate them.
A: Let's think step by step. The last letter of "Gabe" is "e". The last letter of "Ventura" is "a". The last letter of "Woody" is "y". Concatenating them is "eay". The answer is eay. |

Table 16: Exemplars for full chain of thought prompt selected and annotated from Letter (4).

