# OpenReview forum: "Active Prompting with Chain-of-Thought for Large Language Models"
_ICLR.cc/2024/Conference — Submitted to ICLR 2024_

### Official Review · Reviewer_1P1b · 2023-10-29

**Soundness:** 3 good
**Presentation:** 3 good
**Contribution:** 3 good
**Rating:** 6
**Confidence:** 4

**Summary:**

This paper introduces a novel method of incorporating active learning into the exemplar selection process for CoT prompts. Considering that current CoT method relies on a fixed set of human-annotated exemplars, which lacks adaptability for different tasks. The authors propose Active Prompting, selecting the most important and informative samples from the dataset as prompts. Believing that samples with the highest uncertainty are the most helpful, the authors introduce an effective strategy for selecting uncertain samples, along with four metrics for measuring uncertainty.

**Strengths:**

1.	Combining active learning with the selection process of prompt exemplars is a very intriguing and novel perspective.
2.	Experimental results from multiple datasets, coupled with comprehensive analysis, demonstrate the effectiveness of Active Prompt from various angles.
3.	In Section 5.3, the authors' experimental results indicate that uncertain exemplars are transferable, showcasing the superior generalization capability of the method.

**Weaknesses:**

1.	Need for Additional Corpora: One significant advantage of CoT is its ability to leverage the model's generalization capabilities, requiring only a minimal number of task-related samples to teach the model the paradigm for solving tasks. In scenarios without corpora of the same distribution, such as ASDiv, SVAMP, and SingleEq, Active Prompt still needs to capitalize on the model's generalizability. However, it struggles to achieve the same performance boosts as datasets with training corpora like GSM8K and AQuA.
2.	Differences in Uncertainty Measurement Methods: The authors introduced four methods of uncertainty measurement but did not delve deep into their differences. It appears that “Entropy” has better generalizability on datasets like StrategyQA compared to “Disagreement”. However, “Disagreement” outperforms on datasets like SVAMP and CSQA using code-davinci-002, as discussed in Question 2.
3.	Concerns Over Costs: Active Prompt seems to require an additional 1000*k API calls. Given the recommended value of k = 10, an extra 10,000 API calls seems to be a considerably high cost. Additionally, there's the cost associated with extra data annotation, as outlined in Question 3.

**Questions:**

1.	Active learning is an iterative process. However, the method in the article undergoes only a single iteration. The authors also observed that 'the existing annotation of GSM8K is of high quality.' A concerning scenario arises when modifying prompt exemplars causes the model to become uncertain about samples it was previously confident about.
2.	What leads to the performance disparities between Active-Prompt (E) and Active-Prompt (D) across different datasets?
3.	From a cost perspective, does Active Prompt hold any advantages over methods like AutoCoT?
4.	Logically, the larger the value of k, the more accurate the model's uncertainty assessment should be. However, on the SingleEq dataset, a k value of 15 led to a noticeable performance decline. The reason given, 'In careful observation of the dataset, when k > 10, the number of the most uncertain questions is scarce, where confusion is no longer a problem,' is perplexing.
5.	Many current studies have adopted GPT-4 for label generation. For uncertain samples from datasets, can GPT-4 replace human annotation and achieve similar results?

---

> ### Author Response · Authors · 2023-11-19
> **Response to Reviewer 1P1b (1)**
>
> Dear Reviewer 1P1b,
>
> We deeply appreciate your comprehensive review and insightful feedback. Below are our responses to your concerns:
>
> **Q1. Need for Additional Corpora**
>
> From the Table 1, we find that the generalization performance is correlated to the backbone models. If we look into text-davinci-002 and code-davinci-002, we will find the performance boost on GSM8K and AQuA is larger than asdiv, svamp, and singleeq. However, If we look into text-davinci-003 and gpt-3.5-turbo, we will find the performance improvement is similar. For example, with gpt-3.5-turbo, the improvement is around 1.5 for these two scenarios. Therefore, considering the generalization, we can still take the prompts selected by a similar domain / task and then apply them to unseen domains / tasks.
>
> | Model / Dataset | GSM8K | AQuA | Average-1| ASDiv | SVAMP | SingleEq | Average-2 |
> | ----------- | ----------- | ----------- | ----------- | ----------- | ----------- | ----------- |----------- |
> | text-davinci-003 (CoT) | 61.7 | 46.0 | 53.9 | 78.2 | 77.6 | 91.5 | 82.4 |
> | text-davinci-003 (Active-Prompt) | 65.6 | 48.0 | 56.8 | 79.8 | 80.5 | 93.1 | 84.5 |
> | gpt-3.5-turbo (CoT) | 74.2 | 50.0 | 62.1 | 82.5 | 83.8 | 95.0 | 87.1 |
> | gpt-3.5-turbo (Active-Prompt) | 77.1 | 50.0 | 63.6 | 83.6 | 85.5 | 96.0 | 88.4 |
>
>
> **Q2. Differences in Uncertainty Measurement Methods**
>
> Thank you so much for your question! We strongly agree with your observation. In addition, we find entropy-based methods are generally better than disagreement-based methods. For example, across 8 downstream tasks and two models (text-davinci-002 and code-davinci-002), Active-Prompt (E) outperforms Active-Prompt (D) 11 times out of 16. Active-Prompt (D) is better at SVAMP and CSQA. Because the calculation of entropy takes the frequency into consideration, it provides a more accurate estimation for uncertainty.
> For the performance disparities between Active-Prompt (E) and Active-Prompt (D) across different datasets, it is because entropy could capture more fine-grained information to help estimate the uncertainty. Let’s take an example for illustration.
> We query the model 5 times and obtain the follow results:
>
> For question 1, answers A1 = {2, 3, 3, 3, 6}
>
> For question 2: answers A2 = {2, 2, 3, 3, 6}
>
> The disagreement of A1 is 3 and the disagreement of A2 is also 3. However, the entropy is different. The entropy of A1 is 1.37 while the entropy of A2 is 1.52. This explains why entropy-based methods are better than disagreement-based methods across 11 out of 16 tasks. Because of the fine-grained information, entropy could distinguish the uncertainty of different questions better.
>
> **Q3. Costs**
>
> For the cost of selection, compared with selecting questions by humans, our proposed method is more efficient. For a new task, users need to do trials and errors a lot of times which costs a lot of human effort with unstable performance. Even so, the selected questions are still suboptimal.
> Second, as mentioned in Section 3.3, we limit the size of candidate instances to 1,000 which greatly reduce the cost. 1,000 is a good balance between cost and performance. We verified that with more than 1,000 instances, the performance would converge. Doing uncertainty estimation 10 times with a pool of 1,000 questions is acceptable. The cost is smaller than self-consistency, which usually requires 40 times inference, although it is an orthogonal technique and can be complementary to ours. In addition, inspired by the new experimental results requested by reviewer Wb4R, we are excited to find that questions selected by smaller models (e.g., Llama) perform well with larger models (e.g., gpt-3.5-turbo). Considering models like Llama are open-source which does not cause API cost, one may use it (with GPU) to replace black-box API.
>
>
> For the cost of annotation, annotating 8 questions is not costly. Humans could finish it in several minutes. Furthermore, like you suggested, one can use ChatGPT or GPT-4 for annotation, which is much more easier. We tried some examples and found the GPT-4 annotations make a lot of sense and we believe they work well as human’s annotation. We are doing this and will add this experiment very soon (before Nov. 22). Thanks!

---

> ### Author Response · Authors · 2023-11-19
> **Response to Reviewer 1P1b (2)**
>
> **Q4. Iterative Process and Uncertainty**
>
> Thanks for your questions! Conventional active learning is indeed an iterative process, but now some work feels that the iterative method is costly. So, there are also some recent studies that are one-pass [1,2].
> To explore the effects of modification, we conduct more analysis by comparing the changes in terms of confident questions set before and after modification. Here, the confident question set is defined as a set of questions that the model is highly consistency (uncertainty is 0). With gpt-3.5-turbo, using the unmodified CoT, there are 325 questions in the confident question set (set X_1). After modification, with the modified CoT, there are also 325 questions (set X_2). Over 70 % questions in X_1 appear in set X_2. Although the uncertainty has increased in a small number of questions, we find that their uncertainty still remains very low.
>
>
> **Q5. Cost Compared with Auto-CoT**
>
> Auto-CoT applies Sentence-BERT to encode all the questions into vector representation and then adopts k-means clustering to produce k clusters of questions. Active-Prompt does not require CPU and GPU computation because it is purely based on ChatGPT API which costs money. Although there is a clear performance advantage compared with Auto-CoT (as shown in Appendix D), there is no clear advantage compared with Auto-CoT because of the different implementations. Thank you for your question!
>
>
> **Q6. Performance Drop in Figure 2**
>
> Your observation about the accuracy decrease in the SingleEq dataset is astute. We attribute this to potential gaps during the transfer process from GSM8K to SingleEq, likely caused by noise and fluctuation in the inference process. To address this, we conducted additional experiments and updated Figure 2 with more stable and consistent results. Due to the unavailability of the code-davinci-002 model, we switched to the text-davinci-003 model for these experiments. The revised figure now reflects a more consistent increase and convergence in accuracy, which we believe more accurately represents the performance of our method.
>
> Once again, thank you for your valuable feedback, which has significantly contributed to the refinement of our work.
>
>
> [1] Lin, Yong, et al. "Optimal Sample Selection Through Uncertainty Estimation and Its Application in Deep Learning." arXiv preprint arXiv:2309.02476 (2023).
>
> [2] Wang, HaiYing, Rong Zhu, and Ping Ma. "Optimal subsampling for large sample logistic regression." Journal of the American Statistical Association 113.522 (2018): 829-844.

---

> ### Author Response · Authors · 2023-11-20
>
> Dear Review 1P1b,
>
> We sincerely appreciate your insightful feedback once again. In response, we have diligently addressed each of your concerns and queries. Key updates include:
>
> 1. Discussion about the Generalization Performance
> 2. Analysis of the Differences in Uncertainty Measurement Methods
> 3. Discussion about the Cost
> 4. Revisit to the Performance Drop in Figure 2
>
> Would you be able to spare some time to review our revised response and confirm if it adequately addresses your questions? We are eager to engage in further discussions and provide additional clarifications on any new queries you may have.
> Thank you very much!

---

> > ### Comment · Reviewer_1P1b · 2023-11-23
> >
> > Thank you for your detailed and thoughtful responses to the questions.
> >
> > I appreciate the additional experiments and clarifications provided, particularly regarding the generalization performance and the innovative approaches to uncertainty measurement. It would indeed be beneficial for these findings to be included in the final manuscript. Looking forward to seeing the final version of your work!
> >
> > Reviewer 1P1b

---

> > > ### Author Response · Authors · 2023-11-23
> > >
> > > Dear Reviewer 1P1b,
> > >
> > > We would like to thank you again for your effort and positive feedback!
> > >
> > > We are very happy that our response and updated presentation have resolved your concerns. We are also very grateful for your constructive and valuable comments, which helped improve our manuscript a lot and made the paper stronger.
> > >
> > > We really enjoyed the discussion with you. Thank you very much!

---

### Official Review · Reviewer_hZjq · 2023-10-31

**Soundness:** 3 good
**Presentation:** 3 good
**Contribution:** 2 fair
**Rating:** 5
**Confidence:** 4

**Summary:**

The paper proposes a method called Active-Prompt for adapting LLMs to different tasks with specific example chain-of-thought prompts. This method includes determining a subset of examples for each task/dataset based on uncertainty estimation and having human annotators annotate these examples with chain-of-thought reasoning. The authors present four methods for uncertainty estimation: disagreement, entropy, variance and self-confidence, but mainly apply disagreement and entropy based approaches stating that these outperform the rest. The authors compare their approach against baselines (CoT, Self-Consistency, Auto-CoT, and RandomCoT) on different math and commonsense reasoning problems, showing improved performance across different tasks. They also present an ablation study, discussing the effects of few-shot prompts, active selection, annotations, and uncertainty metrics.

**Strengths:**

- Overall the paper is written clearly and proposes an approach for example selection for chain-of-thought prompting. The method uses existing approaches from active learning and shows improvements over baselines.
- The authors evaluate their approach on a range of mathematical and commonsense reasoning tasks, and conduct ablations to understand the effect of different factors.

**Weaknesses:**

- The approach seems to have limited applicability as it requires the existence of either large enough datasets for a particular task or similar task to sample from. The authors also report variations between different annotators, further attesting to the difficulty of the task.
- Some details in the paper are missing. For example, how is the variance based approach applied to textual answers? There are no results presented with the self-confidence approach and only an example is given, etc.

**Questions:**

1- How will the approach generalize to new tasks?
2- How is the variance based approach applied to textual response?
3- In Figure 2, what is the intuition for accuracy decreasing with more number of predicted answers?

---

> ### Author Response · Authors · 2023-11-19
> **Response to Reviewer hZjq**
>
> Dear Reviewer hZjq,
>
> We greatly appreciate your thorough review and insightful feedback. Here are our responses to your concerns:
>
> **Q1. Generalization to New Tasks**
>
> Active-Prompt is designed to be versatile and can be effectively utilized even with a limited number of unlabeled questions. This flexibility allows for efficient generalization to similar tasks. For situations where collecting new questions is feasible, our uncertainty estimation method facilitates the selection of high-quality examples. In cases where question collection is challenging, transferring questions from a similar domain is a viable alternative. Our results in Table 1, showcasing the transferability of selected questions from GSM8K to other datasets like ASDiv, SVAMP, and SingleEq, illustrate this point effectively. This evidence supports the notion that Active-Prompt is not only effctive but also generalizable to new tasks.
>
>
>
> **Q2. Performance of Variance-based and Self-confidence-based Method**
>
> Regarding the variance-based method, its applicability is indeed limited to arithmetic tasks, as it struggles with textual responses. For the self-confidence-based metric, as discussed in Section 5.1 [Effects of Uncertainty Metrics], we observed its tendency towards overconfidence, leading to suboptimal performance. Consequently, our focus shifted to disagreement and entropy metrics, which demonstrated more reliable results. We have clarified these points in our manuscript to ensure a better understanding of our methodological choices. Thanks!
>
>
>
> **Q3. Accuracy Decreasing in Figure 2**
>
> Your observation about the accuracy decrease in the SingleEq dataset is astute. We attribute this to potential gaps during the transfer process from GSM8K to SingleEq, likely caused by noise and fluctuation in the inference process. To address this, we conducted additional experiments and updated Figure 2 with more stable and consistent results. Due to the unavailability of the code-davinci-002 model, we switched to the text-davinci-003 model for these experiments. The revised figure now reflects a more consistent increase and convergence in accuracy, which we believe more accurately represents the performance of our method.
>
> Thank you once again for your constructive feedback, which has been invaluable in enhancing the quality and clarity of our work.

---

> ### Author Response · Authors · 2023-11-20
>
> Dear Review hZjq,
>
> We sincerely appreciate your insightful feedback once again. In response, we have diligently addressed each of your concerns and queries. Key updates include:
>
> 1. Analysis of the Generalization to New Tasks
> 2. Discussion about the Performance of Variance-based and Self-confidence-based Method
> 3. Revisit the Accuracy Decreasing in Figure 2
>
> Would you be able to spare some time to review our revised response and confirm if it adequately addresses your questions? We are eager to engage in further discussions and provide additional clarifications on any new queries you may have.
> Thank you very much!

---

### Official Review · Reviewer_DkmB · 2023-11-01

**Soundness:** 1 poor
**Presentation:** 3 good
**Contribution:** 1 poor
**Rating:** 1
**Confidence:** 5

**Summary:**

This paper proposes an uncertainty-based few-shot example selection/annotation method for LLMs. The motivation is annotating/selecting in-context examples for LLMs could be time-consuming and challenging. Since the few-shot examples significantly influence the downstream performance of LLMs, the authors propose to leverage uncertainty as the indicator to decide which examples should be selected from a large pool of candidate data.  Empirical evaluations demonstrate that the proposed method outperforms previous short and simple chain-of-thought annotations and improve the performance of LLMs.

**Strengths:**

The idea is straightforward and the motivation is clear. The method makes sense.

**Weaknesses:**

1. **Baselines are too weak, leading to a misunderstanding of the effectiveness of the proposed method.** I would like to urge the authors to include more powerful baselines in the experiment rather than hide them. ALL the reviewers are experts in this domain and familiar with the state-of-the-art performance of LLMs on these benchmarks in this domain. In the experiment section, the authors only include the CoT annotations from [1] as the most important baseline. It is widely acknowledged and studied that the complexity (i.e., the length or reasoning steps of the CoT annotations) significantly influences the performance of the LLMs. The annotations from [1] are very simple and short, only including some easy examples as in-context examples. In comparison, if we look at Page 17, the actual annotations from the authors are very long and detailed. Previous work [2] has already shown that by selecting the most complex examples from the training dataset, the performance can be largely improved compared to the original annotations from [1]. For example, by selecting the most complex examples, the performance of ChatGPT (i.e., gpt-3.5-turbo) can easily achieve more than 80% accuracy (without self-consistency) compared to the number 77.1% in Table 1. One may also refer to https://opencompass.org.cn/leaderboard-llm for the performance of LLMs (I acknowledge that the performance of ChatGPT on GSM8K from that website is possibly still underestimated). Without comparison with SOTA's performance, I will try my best to reject this paper. Please do not try to hide the best baselines.

2. **More ablation study is required.** Again, the performance improvement may come from two aspects. The first is selecting the most uncertain examples, and the second is making the CoT annotations longer. The annotations in baseline [1] are much shorter compared to the annotations by the authors. Without the ablation studies on these two aspects, we cannot determine whether the performance improvement truly comes from the author's contribution or just longer CoT annotations.

3. **The method is simple with limited contribution, while performance improvement is not significant.** The method is quite intuitive and can be regarded as an in-context example selection method (followed by annotations). The authors should discuss the relationship with other in-context example selection methods and compare the performance. Existing performance improvement is quite limited. Once more baselines are included, it is very possible that the performance will be surpassed.

[1] Jason Wei, Xuezhi Wang, Dale Schuurmans, Maarten Bosma, Brian Ichter, Fei Xia, Ed H. Chi, Quoc V. Le, and Denny Zhou. Chain-of-thought prompting elicits reasoning in large language models. In NeurIPS, 2022b.

[2] Yao Fu, Hao Peng, Ashish Sabharwal, Peter Clark, and Tushar Khot. Complexity-based prompting for multi-step reasoning. In ICLR 2023

**Questions:**

Please refer to the weakness above. Without my concerns properly addressed (more sufficient and reasonable baselines), I will strongly reject this paper.

---

> ### Author Response · Authors · 2023-11-19
> **Response to Reviewer DkmB (1)**
>
> Dear Reviewer DkmB,
>
> Thank you very much for your comprehensive review and valuable feedback! We address your comments one by one as follows:
>
>
> **Q1. Performance in Table 1**
>
> Thank you for highlighting the discrepancies in the performance of gpt-3.5-turbo on GSM8K as reported in different sources. We acknowledge the variation in results across different papers. For example, according to chain-of-thought hub [3] (https://github.com/FranxYao/chain-of-thought-hub), the performance is 74.9 which is similar to our reported 74.2. But indeed, we find some papers reporting higher performance [4,5], which is about 80.8. We checked Opencompass [6] as well, they reported 78.2. All of these three sources have a much higher performance than our reported 74.2. This leads to an interesting question which motivates us to take a deep investigation.
>
> First of all, we are very certain that the results we reported are not deliberately understated.
> We take a careful look into [4] who reported 80.8 and find they are using a different version of gpt-3.5-turbo. Our initial results were based on gpt-3.5-turbo-0613, while some of the higher performances reported in other studies utilized gpt-3.5-turbo-0301. We conduct experiments with gpt-3.5-turbo-0301 and surprisingly find that the baseline CoT is indeed above 80! We never expect such a situation although there are several papers observing and discussing similar phenomenons [7]. Therefore, we add the results with gpt-3.5-turbo-0301 as well to provide a more comprehensive evaluation. The results are shown below.
>
> | Model / Dataset | GSM8K | AsDiv | SVAMP | SingleEq |
> | ----------- | ----------- | ----------- | ----------- | ----------- |
> | gpt-3.5-turbo-0613 (original CoT)| 74.2 | 82.5 | 83.8 | 95.0 |
> | gpt-3.5-turbo-0613 (active-prompt) | 77.1 | 83.6 | 85.5 | 96.0 |
> | gpt-3.5-turbo-0301 (original CoT) | 80.1 | 86.7 | 82.0 | 91.3 |
> | gpt-3.5-turbo-0301 (active-prompt) | 83.5 | 87.4 | 83.0 | 93.3 |
>
> These results confirm a consistent improvement with our proposed Active-Prompt approach, irrespective of the GPT-3.5 version. We list the results with different versions in the same table for people to reproduce the results. We are grateful for your comment which helps us have a deeper understanding. In addition, the 0613 version is likely based on the continued training from 0301, which reveals a very interesting phenomenon — the forgetting issue of LLMs as training progresses. This has sparked our interest in conducting more in-depth research in the future. Thank you very much!
>
>
> **Q2. Ablation Study of Longer CoT Annotations**
>
> Thank you for suggesting an ablation study to differentiate the impacts of longer CoT annotations from our method. To explore this, we extended the length of the original CoT [1] annotations to an average of 155 words, comparable to our average length of 160 words. Here are the comparative results:
>
>
> | Model / Dataset | GSM8K | AsDiv | SVAMP | SingleEq |
> | ----------- | ----------- | ----------- | ----------- | ----------- |
> | gpt-3.5-turbo (original CoT) | 74.2 | 82.5 | 83.8 | 95.0 |
> | gpt-3.5-turbo (longer CoT) | 69.4 | 69.2 | 70.4 | 83.2 |
> | gpt-3.5-turbo (active-prompt) | 77.1 | 83.6 | 85.5 | 96.0 |
>
> Our findings show that merely increasing the length of CoT annotations does not lead to improved performance, and in some cases, even reduces it. In contrast, our Active-Prompt method consistently demonstrates superior performance. This suggests that the selection of questions, rather than their length, contributes significantly to the improved results. Our approach effectively identifies and utilizes more informative examples for annotations.

---

> > ### Author Response · Authors · 2023-11-19
> > **Response to Reviewer DkmB (2)**
> >
> > **Q3. Performance improvement and more baselines**
> >
> > We appreciate your emphasis on comparing our method with diverse baselines. We have included comparisons with both diversity-based (Auto-CoT [8]) and complexity-based (Complex-CoT [2]) methods.
> > First, the comparison with Auto-CoT is as follows.
> >
> > | Model / Dataset | GSM8K | MultiArith | AddSub |
> > | ----------- | ----------- | ----------- | ----------- |
> > | Complex-CoT | 62.8 | 93.2 | 91.9 |
> > | Active-Prompt (D) | 67.0 | 95.5 | 93.2 |
> >
> > Because Auto-CoT only reported the results on GSM8K, MultiArith, and AddSub with code-davinci-002 without self-consistency, we compare our method with it on these three datasets in the same setting, and we find Active-Prompt outperforms Auto-CoT by a large margin.
> >
> > Second, we test the performance of Complex-CoT and compare it with Active-Prompt, and the results are shown in the following table. All the results are based on gpt-3.5-turbo-0613. The prompts for Complex-CoT are directly taken from their paper.
> >
> > | Model / Dataset | GSM8K | AsDiv | SVAMP | SingleEq |
> > | ----------- | ----------- | ----------- | ----------- | ----------- |
> > | Complex-CoT | 76.3 | 82.4 | 79.9 | 93.3 |
> > | Active-Prompt (D) | 77.1 | 83.6 | 85.5 | 96.0 |
> >
> > We find that Active-Prompt outperforms Complex-CoT across all tasks, demonstrating the effectiveness of uncertainty-based methods.
> >
> > Note that diversity, complexity, and uncertainty are useful for selecting the most informative questions, and they are complementary. We consider the combination of them as an important future direction.
> >
> >
> >
> >
> >
> > [1] Jason Wei, Xuezhi Wang, Dale Schuurmans, Maarten Bosma, Brian Ichter, Fei Xia, Ed H. Chi, Quoc V. Le, and Denny Zhou. Chain-of-thought prompting elicits reasoning in large language models. In NeurIPS, 2022b.
> > [2] Yao Fu, Hao Peng, Ashish Sabharwal, Peter Clark, and Tushar Khot. Complexity-based prompting for multi-step reasoning. In ICLR 2023
> >
> > [3] Yao Fu, Litu Ou, Mingyu Chen, Yuhao Wan, Hao Peng, and Tushar Khot. Chain-of-Thought Hub: A Continuous Effort to Measure Large Language Models' Reasoning Performance. arXiv preprint arXiv:2305.17306 (2023).
> >
> > [4] Yuxi Xie, Kenji Kawaguchi, Yiran Zhao, Xu Zhao, Min-Yen Kan, Junxian He, and Qizhe Xie. Self-Evaluation Guided Beam Search for Reasoning. In NeurIPS, 2023.
> >
> > [5] Zhibin Gou, Zhihong Shao, Yeyun Gong, Yujiu Yang, Minlie Huang, Nan Duan, and Weizhu Chen. ToRA: A Tool-Integrated Reasoning Agent for Mathematical Problem Solving. arXiv preprint arXiv:2309.17452 (2023).
> >
> > [6] OpenCompass Contributors, OpenCompass: A Universal Evaluation Platform for Foundation Models.
> >
> > [7] Lingjiao Chen, Matei Zaharia, and James Zou. "How is ChatGPT's behavior changing over time?." arXiv preprint arXiv:2307.09009 (2023).
> >
> > [8] Zhuosheng Zhang, Aston Zhang, Mu Li, and Alex Smola. "Automatic chain of thought prompting in large language models." in ICLR 2023.

---

> ### Author Response · Authors · 2023-11-20
>
> Dear Reviewer DkmB,
>
> We sincerely appreciate your insightful feedback once again. In response, we have diligently addressed each of your concerns and queries. Key updates include:
>
> 1. A deep analysis of the performance in Table 1. We explain that our reported results are not deliberately understated and confirm a consistent improvement with Active-Prompt.
> 2. Ablation Study of Longer CoT Annotations
> 3. Comparison and discussion with more baselines
>
>
> Would you be able to spare some time to review our revised response and confirm if it adequately addresses your questions? We are eager to engage in further discussions and provide additional clarifications on any new queries you may have.
>
> Thank you very much for your time and consideration.

---

### Official Review · Reviewer_Wb4R · 2023-11-02

**Soundness:** 2 fair
**Presentation:** 3 good
**Contribution:** 2 fair
**Rating:** 3
**Confidence:** 4

**Summary:**

This paper proposes a new few-shot prompt construction method for LLMs that is inspired by active learning. Assuming access to some training instances, the paper proposes to include as in-context learning examples those that the model is most uncertain about. If these instances do not come with labels, they are manually annotated. This is achieved by testing the model on (a subset of) the training data and finding the instances that yield the highest uncertainty measured by (1) entropy or (2) disagreement. (1) and (2) lead to two variants of the proposed model.

Experiments are conducted on reasoning and QA tasks, with the OpenAI models. The analysis is extensive and insightful. Overall the paper presents an interesting and intuitive idea, and the execution is great. However, I have three major concerns that lead me to vote for a rejection (details below). I am happy to revisit this if the authors can address my concerns.

**Strengths:**

- Combining active learning with prompt construction is interesting and novel to me
- With the extensive experiments and analysis, the execution is definitely above average
- Writing is clear

**Weaknesses:**

- [Major] An important and very relevant baseline is missing: https://arxiv.org/abs/2210.00720. Their method is very similar to Active Prompt and simply selects the longest training instances. I would be curious to see how it compares to this work.
- [Major] One can imagine that if the model is reasonably good, the demonstrations selected by Active-Prompt will be more useful. I wonder whether this is still the case for “weaker” models. If the model does not know too much about the task, will the prompts selected by its uncertainty still be useful? This can be tested out by trying Active-Prompt on, e.g., one of the smaller Llama models.
- [Major] The conclusion drawn on the transferability of prompts found by Active-Prompt in 5.3 needs more evidence. All the models tested are from the GPT-3 family, which are finetuned from the same base model. It is unclear whether, e.g, the prompts found by GPT-3.5 perform well for Llama. This concern is important since it directly determines how useful Active-Prompt is in practice. If the prompts do not transfer across different model families, it will have a huge overhead annotating a new set of instances for a different model. Besides, it makes it impossible to do fair comparisons among models controlling the prompts. I suggest adding an experiment studying the transferability between GPT and Llama models.
- To draw conclusions on the transferability of the prompts, Table 3 should compare, e.g., CD-002->TD-002 (SC) with TD-002->TD-002 (SC), instead of the non-Active-Prompt baseline.
- Some of the wordings are confusing, even misleading. Please see the details below.
- A clear limitation of Active-Prompt is the high cost associated with doing inference runs over the training set. A discussion about this would be nice.

**Questions:**

Below are comments instead of questions, and the authors do not need to answer them.

- The end of page 2, $q_i$ is overloaded, and it is hard to distinguish between instances from training and test data. Adding a superscript or changing the letter could help.
- Above Eq. 2: is “Arabic answers” a typo? Do the authors mean “arithmetic” instead?
- Below Eq.3, $P_{\theta}$ is a distribution, not a random variable.

---

> ### Author Response · Authors · 2023-11-19
> **Response to Reviewer Wb4R (1)**
>
> Dear Reviewer Wb4R,
>
> We are grateful for your comprehensive and insightful review. Your feedback has been instrumental in refining our manuscript. Below, we address each of your points in detail:
>
>
> **Q1. Performance of Complex-CoT**
>
> We sincerely appreciate your suggestion to compare the performance of Complex-CoT with our proposed Active-Prompt methodology. As recommended, we conducted a detailed evaluation using the GPT-3.5-turbo-0613 model. The prompts for Complex-CoT were sourced directly from the original paper to maintain consistency in comparison.
>
>
> | Model / Dataset | GSM8K | AsDiv | SVAMP | SingleEq | Average |
> | ----------- | ----------- | ----------- | ----------- | ----------- | ----------- |
> | Complex-CoT | 76.3 | 82.4 | 79.9 | 93.3 | 82.97 |
> | Active-Prompt (D) | 77.1 | 83.6 | 85.5 | 96.0 | 85.5 |
>
> We find that Active-Prompt consistently outperforms Complex-CoT across all tasks. In addition, we can combine the uncertainty and the complexity together to achieve a better performance, and we look forward to exploring the combination in future work. We have updated our draft with these new results in Appendix E. Thanks very much for your constructive suggestion!
>
>
>
> **Q2. Performance of Weaker Models**
>
> We acknowledge the importance of evaluating performance across models of varying capabilities. Therefore, we conduct new experiments with Llama2-13b-chat and Lllama2-70b-chat. The results are shown as follows.
>
> | Model / Dataset | GSM8K | AsDiv | SVAMP | SingleEq | Average |
> | ----------- | ----------- | ----------- | ----------- | ----------- | ----------- |
> | CoT (Llama2-13b-chat) | 29.1 | 57.4 | 45.7 | 75.4 | 51.9 |
> | Active-Prompt (Llama2-13b-chat) | 27.2 | 53.9 | 47.4 | 68.2 | 49.2 |
> | CoT (Llama2-70b-chat) | 54.8 | 73.2 | 77.4 | 84.6 | 72.5 |
> | Active-Prompt (Llama2-70b-chat) | 56.9 | 74.9 | 82.5 | 83.2 | 74.4 |
> | CoT (gpt-3.5-turbo-0613) | 74.2 | 82.5 | 83.8 | 95.0 | 83.8 |
> | Active-Prompt (gpt-3.5-turbo-0613) | 77.1 | 83.6 | 85.5 | 96.0 | 85.6 |
>
>
> Firstly, because smaller models do not have a good chain-of-thought ability, it is difficult for Llama2-13b-chat to understand the chain-of-thought prompting, with poor performance of both CoT and Active-Prompt. Especially, we find that the more complex the prompt, the poorer performance. However, with Llama2-70b model, our proposed Active-Prompt outperforms CoT by a large margin, demonstrating this method is still useful for weaker models.
>
> Note that we are using the instruction-tuned version of Llama2-70b in all our experiments (i.e., Lllama2-70b-chat) because it is able to understand complex chain-of-thought prompting and follow human instructions.
>
>
>
> **Q3. Transferability between GPT and Llama models**
>
> Thanks for your suggestion! Your suggestion prompted us to investigate the transferability between GPT and Llama models. Because smaller Llama models perform poorly on reasoning tasks, we conduct experiments with Llama2-70b-chat.
> We conduct two types of experiments: (1) select questions by gpt-3.5-turbo and infer by Llama2-70b-chat and (2) select questions by Llama2-70b-chat and infer by gpt-3.5-turbo. Note that we are using the 0613 version of gpt-3.5-turbo.
> The results are shown as follows:
>
> | Model / Dataset | GSM8K | AsDiv | SVAMP | SingleEq | Average |
> | ----------- | ----------- | ----------- | ----------- | ----------- | ----------- |
> | gpt-3.5-turbo  | 74.2 | 82.5 | 83.8 | 95.0 | 83.8 |
> | gpt-3.5-turbo -> gpt-3.5-turbo  | 77.1 | 83.6 | 85.5 | 96.0 | 85.6 |
> | Llama2-70b-chat -> gpt-3.5-turbo  | 78.7 | 85.9 | 83.1 | 91.5 | 84.8 |
> | Llama2-70b-chat | 54.8 | 73.2 | 77.4 | 84.6 | 72.5 |
> | Llama2-70b-chat -> Llama2-70b-chat | 56.9 | 74.9 | 82.5 | 83.2 | 74.4 |
> | gpt-3.5-turbo -> Llama2-70b-chat | 58.9 | 74.7 | 81.2 | 86.0 | 76.0 |
>
> The model before the arrow denotes the model for actively selecting questions while the model after the arrow denotes the model for inference. The results demonstrate the feasibility of selecting questions with smaller models (Llama2-70b-chat) and applying the selected questions to larger models (gpt-3.5-turbo). In addition, selecting questions with larger models and applying them to smaller models has better performance.

---

> > ### Author Response · Authors · 2023-11-19
> > **Response to Reviewer Wb4R (2)**
> >
> > **Q4. Transferability Results in Table 3**
> >
> > Thanks for your suggestions! We add the experimental results of TD-003 -> TD-003 (CoT) in Table 3. TD-003-> TD-003 (CoT) outperforms CD-002 -> TD-003 (CoT) by a large margin (on average 2%). Due to the high inference cost of SC, we did not include the results of TD-002 -> TD-002 (SC). In the transferability analysis, our original intention is not to prove that transfer is better than non-transfer, but to demonstrate that the method of selecting questions through transfer is superior to not selecting them. Transfer will certainly lead to a decrease in results, and we have already updated these results in the paper. Thank you!
> >
> > | Model / Dataset | GSM8K | CSQA | Letter (4) | Average |
> > | ----------- | ----------- | ----------- | ----------- | ----------- |
> > | TD-002 (CoT) | 46.9 | 73.5 | 56.6 | 59.0 |
> > | TD-002 -> TD-002 (CoT) | 48.4 | 74.7 | 57.7 | 60.3 |
> > | TD-002 (SC) | 58.2 | 72.9 | 57.6 | 62.9 |
> > | CD-002 -> TD-002 (SC) | 73.2 | 76.6 | 67.7 | 72.5 |
> > | TD-003 (CoT) | 61.7 | 76.2 | 70.2 | 69.4 |
> > | CD-002 -> TD-003 (CoT) | 65.6 | 78.9 | 71.2 | 71.9 |
> > | TD-003 -> TD-003 (CoT) | 67.2 | 80.8 | 73.7 | 73.9 |
> >
> >
> > **Q5. the high cost associated with doing inference runs over the training set**
> >
> > Thanks for your question!
> > Compared with selecting questions by humans, our proposed method is more efficient. For a new task, users need to do trials and errors a lot of times which costs a lot of human effort with unstable performance. Even so, the selected questions are still suboptimal.
> > Second, as mentioned in Section 3.3, we limit the size of candidate instances to 1,000 which greatly reduces the cost. 1,000 is a good balance between cost and performance. We verified that with more than 1,000 instances, the performance would converge. Doing uncertainty estimation 10 times with a pool of 1,000 questions is acceptable. The cost is smaller than self-consistency, which usually requires 40 times inference, although it is an orthogonal technique and can be complementary to ours.
> > In addition, inspired by the new experimental results requested by you, we are excited to find that questions selected by smaller models (e.g., Llama) perform well with larger models (e.g., gpt-3.5-turbo). Considering models like Llama are open-source which does not cause API cost, one may use it (with GPU) to replace black-box API.  We included this discussion into Section H.
> > Thank you very much for your suggestions!
> >
> >
> > **Q6. Wordings**
> >
> > Thank you for pointing out the need for clearer wordings. We have thoroughly revised the manuscript to address this following your suggestions, with significant changes highlighted in **blue** for ease of review.
> >
> > Thank you again for your valuable suggestions! We have included all these new results in our revised version.

---

> ### Author Response · Authors · 2023-11-20
>
> Dear Reviewer Wb4R,
>
> We are deeply grateful for your valuable insights during the initial review of our ICLR submission. Based on your feedback, we have meticulously revised our paper. Key updates include:
>
> 1. A comparative analysis with Complex-CoT.
> 2. Experiments incorporating weaker models.
> 3. An exploration of transferability between GPT and Llama models.
> 4. Enhanced clarity in our writing to prevent potential misunderstandings.
>
> Would you be able to spare some time to review our revised response and confirm if it adequately addresses your questions? We are eager to engage in further discussions and provide additional clarifications on any new queries you may have.
>
> Thank you very much for your time and consideration.

---

> > ### Comment · Reviewer_Wb4R · 2023-11-22
> >
> > I appreciate the authors' effort conducting the new experiments. Some of my concerns have been addressed, but many are confirmed:
> > - Active Prompting underperforms when used with "weaker" models.
> > - My interpretation of the new prompt transferability experiment is that it shows mixed results. For example, with prompts selected by Llama2-70b-chat, GPT-3.5-turbo underperforms the CoT baseline on 2 out of the 5 datasets.
> > - I disagree with the authors' argument on the cost. First, when there is labeled training data, baselines like [1] does not "require selecting examples by human"; when there is no labeled training data, both Active Prompting and this baseline require human efforts. Second, being able to select prompts by running open-source models like Llama does not make things cheaper, if the cost of the computational resources is considered (e.g., cost of the GPU, cloud compute, and electricity). In fact, I would not be surprised if prompting Llama2-70b is more expensive than using GPT-3 API.
> >
> > I have a clarifying question. The author stated that "The prompts for Complex-CoT were sourced directly from the original paper to maintain consistency in comparison." However, I could not find results on some of the datasets (e.g., AsDIV, SVAMP, SingleEq) from [1]. Can the authors clarify how they were able to reuse the prompts from [1] on these datasets?
> >
> > [1]  https://arxiv.org/abs/2210.00720

---

> ### Author Response · Authors · 2023-11-22
>
> Dear Reviewer Wb4R,
>
> Thank you very much for your insightful feedback!
>
> - For the performance with “weaker” models, it is reasonable to see the results may not match those of “stronger” models. As highlighted in prior research [1], complex reasoning tasks such as GSM8K have been challenging for language models, often necessitating larger model sizes. Given that understanding the Chain of Thought (CoT) is difficult for "weaker" models, the observed results align with expectations. It is important to note that our original manuscript did not assert the applicability of our method to weaker models; our primary focus is on enhancing performance in larger models. The notable performance of our method with the Llama-70b model underscores its effectiveness in open-source models as well with an improvement of 1.9%.
>
> - In terms of the prompts for SVAMP and SingleEq, they are derived from the GSM8K dataset and directly utilized in inference. The less favorable outcomes in these two datasets likely stem from the limited generalization capability of smaller model selections. Nonetheless, the substantial improvement on GSM8K (4.5%) demonstrates our method's strong in-domain transferability, although its out-of-domain transferability merits further exploration. We appreciate your suggestion and will incorporate these findings and modify our claims to make them more precise in our manuscript.
>
> - Regarding the cost, we categorize it into two segments: the cost of question selection and the cost of CoT and answer annotation. Initially, in the absence of Active-Prompt and Complex-CoT, manual question selection is required, which constitutes the "human effort" we previously referenced. As for the cost of CoT and answer annotation, nearly all methods, including the baseline CoT [2], entail some cost, so our approach does not introduce additional expense in this regard. Moreover, we are trading a minimal cost in question selection for notable performance gains. Several enhancements in CoT also involve similar trade-offs, such as the incorporation of self-consistency [3] and clustering [4]. For your reference, the cost for selecting from over 1000 questions with gpt-3.5-turbo is around 5 US Dollars after inferencing 5 times for each question. We will report the details of the cost in the Appendix of our manuscript. Thanks!
>
> - For the prompts of the three datasets (AsDIV, SVAMP, SingleEq), we utilize the same ones as for GSM8K, consistent with the approach in [2]. This practice of transferring GSM8K prompts to these datasets is a standard approach, widely adopted in subsequent CoT research, including ours. The Complex-CoT's prompts are available at https://github.com/FranxYao/chain-of-thought-hub/blob/main/gsm8k/lib_prompt/prompt_hardest.txt
>
> We are truly grateful for your time and valuable contributions to this dialogue. Your constructive engagement has significantly enriched our understanding and focus on the model's additional features, which we believe do not detract from our method's primary experimental performance enhancements. We will incorporate the outcomes of these discussions into our revised manuscript and refine our statements regarding transferability for greater precision. Thank you!
>
> [1] Wei, Jason, et al. "Emergent abilities of large language models." Transactions on Machine Learning Research (TMLR), 2022
>
> [2] Wei, Jason, et al. "Chain-of-thought prompting elicits reasoning in large language models." Advances in Neural Information Processing Systems 35 (2022): 24824-24837.
>
> [3] Wang, Xuezhi, et al. "Self-consistency improves chain of thought reasoning in language models." The Eleventh International Conference on Learning Representations. ICLR 2023
>
> [4] Zhang, Zhuosheng, et al. "Automatic chain of thought prompting in large language models."The Eleventh International Conference on Learning Representations. ICLR 2023

---

> > ### Comment · Reviewer_Wb4R · 2023-11-23
> >
> > Thanks for following up and addressing my questions. The submission has improved, primarily by including the Complex-CoT baseline. However, my other concerns have not been addressed:
> > - On Active Prompting's suboptimal performance with weaker models: a different way to interpret this finding is that the proposed approach will be less useful for more challenging tasks, where even the larger models underperforms.
> > - On the cost: sorry for the confusing wording I used---by cost I meant not only in terms of the dollar amount, but also the efficiency, and the additional overhead of making fair comparisons among different models. Based on the results, I find it hard to justify running an LLM over 1K training samples compared to, e.g., simply selecting the longer examples or even randomly drawing the demonstrations.
> > - On the transferability: it seems that the authors and I both agree that this is a limitation of the proposed approach.
> >
> > I have also read reviews and discussion in other thread. I decide to keep my initial score.

---

### Author Response · Authors · 2023-11-21
**General Response**

To all reviewers:

We would like to thank all reviewers for your insightful comments and valuable feedback. We are excited that you found our proposed approach to be **novel, interesting and intriguing** [Reviewers Wb4R, 1P1b] with **clear motivation** [Reviewer DkmB]; our experiments to be **extensive with great execution** [Reviewer Wb4R], and **comprehensive** [Reviewer 1P1b]; our draft to be quite **clear and well written** [Reviewers Wb4R, hZjq]. In addition, the transferability displayed in our paper shows the superior generalization capability of the method [Reviewer 1P1b].

We also appreciate many helpful suggestions, based on which we have improved our manuscript and uploaded a revised version of our submission, with major changes highlighted in **blue**. The main changes are:
In Appendix E, we added the comparison with Complex-CoT.
In Appendix F, we reported the performance of weaker models (i.e., Llama).
In Appendix G, we added new experiments about the transferability between GPT and Llama models.
In Appendix H, we discussed the costs of our method.
In addition, we also addressed typos and improved the wording in the updated manuscript. Thank you for your very constructive suggestions to help improve our manuscript's quality!

We would again like to thank all reviewers for their time and effort, and we hope that our changes adequately address all concerns. We are very happy to have further discussions and address any remaining points during this phase.

---

### Meta-Review · Area_Chair_ScGW · 2023-12-13

**Metareview:**

The paper proposes a few-shot prompting method inspired by active learning. It leverages uncertainty as the indicator to decide which examples should be selected as in-context examples from a training data. While the general idea of active learning for ICL is interesting, the reviewers identified some key issues with the work. These include: (a) suboptimal performance with weaker models, (b) computation cost involved, (c) weaker baselines and insufficient ablations, and (d) limited applicability. They remain unconvinced and I agree with most of their concerns.

**Justification For Why Not Higher Score:**

NA

**Justification For Why Not Lower Score:**

NA

---

### Decision · Program_Chairs · 2024-01-16

Reject